

# 'Rolling' stoneflies (Insecta: Plecoptera) from mid-Cretaceous Burmese amber

Pavel Sroka[1], Arnold H. Staniczek[2] and Boris C. Kondratieff[3]

[1] Biology Centre of the Czech Academy of Sciences, Institute of Entomology, České Budějovice, Czech Republic
[2] Department of Entomology, Stuttgart State Museum of Natural History, Stuttgart, Germany
[3] Department of Bioagricultural Sciences and Pest Management, Colorado State University, Fort Collins, CO, USA

## ABSTRACT

This contribution describes seven new species of fossil stoneflies from Cretaceous Burmese amber, all of which are dedicated to present and past members of the Rolling Stones. Two species—*Petroperla mickjaggeri* gen. nov. sp. nov. and *Lapisperla keithrichardsi* gen. nov. sp. nov.—are placed in a new family Petroperlidae within the stemline of Systellognatha. The first Cretaceous larval specimen of Acroneuriinae, *Electroneuria ronwoodi* gen. nov. sp. nov., is also described along with another four new species that are placed within the Acroneuriinae genus *Largusoperla* Chen et al., 2018: *Largusoperla charliewattsi* sp. nov., *Largusoperla brianjonesi* sp. nov., *Largusoperla micktaylori* sp. nov., and *Largusoperla billwymani* sp. nov. Additional specimens of Acroneuriinae are described without formal assignment to new species due to insufficient preservation. Implications for stonefly phylogeny and palaeobiogeography are discussed.

## INTRODUCTION

Stoneflies (Plecoptera) is a small insect order with approximately 3,700 extant and 250 fossil species currently described (*DeWalt et al., 2018*). Extant stonefly diversity is highest in the temperate regions of the Northern Hemisphere (*Fochetti & De Figueroa, 2008*), a pattern probably consistently followed throughout their history (*Sinitshenkova, 1997*). Stoneflies in general have aquatic larvae and usually prefer cold streams with relatively high amount of dissolved oxygen (*Zwick, 1973*; *Hynes, 1976*), although there are both extant and extinct Jurassic taxa known from hypotrophic lakes (*Brittain, 1990*; *Sinitshenkova & Zherikin, 1996*).

Plecoptera is considered either as sister group to all other Neoptera or as one of the basal clades within Polyneoptera (*Zwick, 2009*; *Misof et al., 2014*; *Wang et al., 2016*). Phylogeny of individual lineages within the order was analysed by *Zwick (1973*, *1980)*. *Sinitshenkova (1987)* presented a phylogeny that also included fossil taxa. However, partly due to the limited number of characters visible in some fossil taxa, her topology is not supported by reliable apomorphies of individual clades. The most recent and widely

Corresponding author
Arnold H. Staniczek,
arnold.staniczek@smns-bw.de

accepted phylogeny of Plecoptera was published by *Zwick (2000)* based on morphological characters of extant species only.

The first molecular phylogeny was published by *Thomas et al. (2000)* using 18S gene only. A more complex study using six molecular markers (12S, 16S, 18S, 28S, COII, H3) and morphological data was performed by *Terry (2003)*. Results of this study were largely congruent with *Zwick (2000)*, recovering all major groups, i.e. Antarctoperlaria, Arctoperlaria, Systellognatha, Euholognatha (without *Megaleuctra*), and Perloidea as monophyletic. Recently, *Chen et al. (2018)* proposed a molecular phylogeny of Systellognatha inferred from mitochondrial genome sequences.

Stoneflies are relatively rare in the fossil record, which might be related to their delicate bodies and their preferred lotic habitat preventing fossilization. Stoneflies date back to the Carboniferous, with the alleged stem-Plecoptera *Gulou carpenteri Béthoux et al., 2011* from the Pennsylvanian of China (*Béthoux et al., 2011*), although there is only indirect evidence for its placement within Plecoptera. Numerous stonefly fossils attributed to the crown group are known from the Permian, with the diversity culminating in the Jurassic and declining in the Early Cretaceous (*Sinitshenkova, 1997*; *Liu & Ren, 2006*).

The best-preserved fossil specimens are found in amber, allowing the observation of even minute morphological details. However, amber inclusions of stoneflies are in general very rare. Only one species attributed to the Perlidae subfamily Acroneuriinae (*Stark & Lenz, 1992*) was described from Miocene Dominican amber. Three species of Capniidae and Leuctridae were recognized in Rovno amber (*Sinitshenkova, 2009*), and 19 species out of five families (Taeniopterygidae, Leuctridae, Nemouridae, Perlidae, Perlodidae) are described from Eocene Baltic amber, all but two of which are attributed to extant genera (*Caruso & Wichard, 2010*, *2011*; *Chen, 2018a*). The occurrence of Plecoptera in Upper Cretaceous Siberian amber is also known (*Wichard & Weitschat, 1996*). Only recently, *Chen, Wang & Du (2018)* and *Chen (2018b*, *2018c)* reported the first Plecoptera from mid-Cretaceous Burmese amber with the descriptions of the new genera *Largusoperla Chen et al., 2018* (Perlidae: Acroneuriinae) including four species, and *Pinguisoperla* Chen, 2018, including a single species *Pinguisoperla yangzhouensis*.

In this study, we add four new species of *Largusoperla* and describe the first larval specimen of stoneflies in Burmese amber. More importantly, we report two new genera within a new family of fossil stoneflies, which probably represent stem group representatives of Systellognatha. We discuss their systematic affinities and evaluate possible implications for Plecoptera phylogeny. We also provide novel insights into several aspects of the palaeocology and palaeobiogeography of stoneflies.

## MATERIAL AND METHODS

### Specimens

All pieces of fossil-bearing Burmese amber originate from the type locality Hukawng Valley, Kachin State, Myanmar. The exact outcrop among the various amber mines in this valley is unknown, because the specimens were acquired from traders. A review of these amber deposits and relative geological history is available in *Zherikhin & Ross (2000)*,

*Grimaldi, Engel & Nascimbene (2002)*, and *Ross et al. (2010)*. UePb zircon dating (*Shi et al., 2012*) constrained this amber to a maximum age of 98.79 ± 0.62 Ma, which is equivalent to the earliest Cenomanian (*Gradstein, Ogg & Smith, 2004*).

## Imaging

The material was studied under a Leica M205 C (Leica Corporation, Wetzlar, Germany) and Olympus SZX7 (Olympus Corporation, Tokyo, Japan) stereomicroscope. Leica Z16 APO Macroscope with Leica Application Suite Version 3.1.8 and Helicon Focus Pro was used to obtain stacked photographs with extended depth of field. Photographs were sharpened and adjusted in contrast and tonality in Adobe Photoshop™ version CS6 (Adobe Systems Incorporated, San Jose, CA, USA).

High-resolution μ-CT scanning with a Bruker Skyscan 1272 tomograph was performed on specimen SMNS BU-79, but yielded no sufficient contrast to produce any useful tomographic image.

## Terminology

Abbreviations for wing veins used throughout the text follow *Béthoux (2005)*: C, costa; ScP, subcosta posterior; R, radius; RA, radius anterior; RP, radius posterior; M, media; MA, media anterior; MP, media posterior; Cu, cubitus; CuA, cubitus anterior; CuP, cubitus posterior; AA, analis anterior; arc, arculus (secondarily strenghtened cross vein between M and Cu).

All type specimens examined are housed in the State Museum of Natural History, Stuttgart, Germany (SMNS) and catalogue numbers are specified for individual species below.

When discussing the affinities of individual taxa, we refer to the phylogenetic system of Plecoptera proposed by *Zwick (2000)*.

## Nomenclature

The electronic version of this article in portable document format will represent a published work according to the International Commission on Zoological Nomenclature (ICZN), and hence the new names contained in the electronic version are effectively published under that Code from the electronic edition alone. This published work and the nomenclatural acts it contains have been registered in ZooBank, the online registration system for the ICZN. The ZooBank LSIDs (Life Science Identifiers) can be resolved and the associated information viewed through any standard web browser by appending the LSID to the prefix http://zoobank.org/. The LSID for this publication is: urn:lsid:zoobank.org:pub:486E9A01-EF59-41D7-B001-4AD6D7FBB11F. The online version of this work is archived and available from the following digital repositories: PeerJ, PubMed Central, and CLOCKSS.

## Etymology of 'Rolling' stoneflies

Burmese amber is one of the oldest resins with insect inclusions, and stoneflies are one of the oldest pterygote lineages. What lies closer at hand than to link fossil stoneflies in ancient stones with the Rolling Stones and to name the new species after the members

of the oldest and greatest Rock 'n' Roll Band in the world. The discerning reader will notice that the new family and genera are named after 'the Stones,' and all present and former members of the Rolling Stones are honoured with their own species.

## SYSTEMATIC PALAEONTOLOGY

Class Insecta Linnaeus, 1758

Subclass Pterygota Lang, 1888

Order Plecoptera Burmeister, 1839

Suborder Arctoperlaria Zwick, 1973

Infraorder Systellognatha Zwick, 2000

**Petroperlidae, fam. nov.**

urn:lsid:zoobank.org:act:3F9EB209-A5DD-49C9-A5D6-7952F05F94A4

*Type genus. Petroperla* gen. nov.

**Diagnosis.** Glossae and paraglossae of approximately same size (plesiomorphy of Systellognatha), rather stout labial palps (plesiomorphy of Systellognatha), short first tarsomere (apical tarsomere 3.5× longer than first tarsomere) (apomorphy of Systellognatha); euplantulae present on tarsi (plesiomorphy of Systellognatha); setose arolium (apomorphy of Systellognatha); forewing with numerous crossveins in costal field (plesiomorphy of Systellognatha); vein RA almost reaching wing apex (apomorphy of Petroperlidae), proximal origin of vein RP (just distal to 1/4 of wing length) (plesiomorphy of Systellognatha).

*Petroperla* **gen. nov.**

urn:lsid:zoobank.org:act:68557BD3-F4E1-43BB-A10B-68D796971769

**Type species.** *Petroperla mickjaggeri*, gen. et sp. nov.

**Diagnosis.** By monotypy, as for the type species.

**Etymology.** The first part of the compound noun refers to the Rolling Stones and is derived from Latin 'petra,' meaning 'stone,' the second part 'perla,' refers to the stonefly genus *Perla*.

*Petroperla mickjaggeri* **sp. nov.** (Figs. 1 and 2)

urn:lsid:zoobank.org:act:838EDFF8-BD85-4F83-88F6-87E38701A941

**Diagnosis.** Thoracic gill remnants absent, abdominal segments not extended posterolaterally.

**Etymology.** The name refers to Sir Mick Jagger, founding member, harmonica player, and lead singer of the Rolling Stones.

**Material.** Holotype specimen: SMNS BU-79, female.

**Description.**
Body length 6.2 mm.

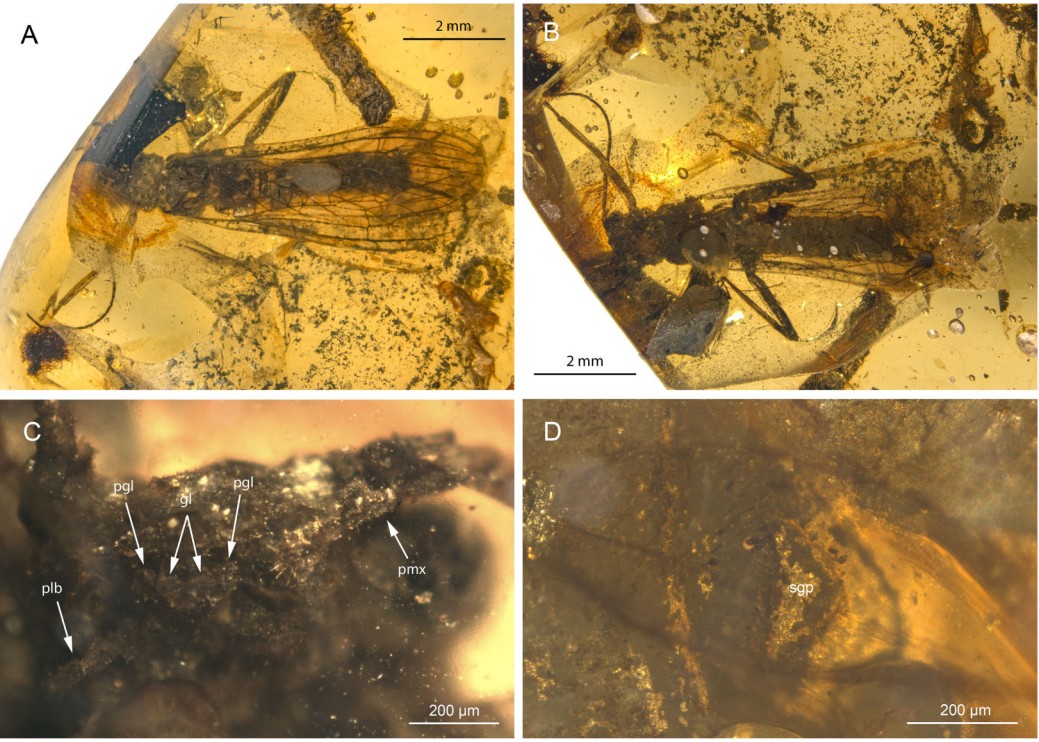

**Figure 1** *Petroperla mickjaggeri* **gen. nov. sp. nov., holotype SMNS BU-79, photographs.** (A) Dorsal view. (B) Ventral view. (C) Head in frontoventral view. Abbreviations: gl, glossa; pgl, paraglossa; pmx, maxillary palp; plb, labial palp. (D) Ventral tip of abdomen with subgenital plate: sgp.

*Head.* Colour greyish, original cuticular colouration not preserved. All three ocelli fully developed. Median ocellus slightly subequal in size to lateral ocelli. Antennae 3.6 mm long (approximately 0.6× body length). Individual segments covered with short hair-like setae. Antennal segments in middle third of antenna approximately 2.2× longer than wide. Maxillary palps five-segmented, two basal palpomeres short, three distal palpomeres longer, of approximately same length. Labium with glossae and paraglossae of approximately same size, labial palps rather stout, three-segmented (Figs. 1C and 2A). Two basal palpomeres short, apical palpomere longer. Other mouthparts not observable in detail.

*Thorax.* Prothorax approximately quadrangular. Colour greyish, original cuticular colouration not preserved (Figs. 1A and 1B).

Forewings (Figs. 2E and 2F): length 6.5 mm, width 2 mm; costal field with numerous crossveins (12 crossveins visible on right forewing and 17 on left forewing, including area distal to ScP, badly visible due to preservation); ScP reaching RA just distally to 1/2 of wing length; RA simple, almost reaching wing apex; single crossvein between RA and RP; RP originating from R just distally to 1/4 of wing length; RP with two distal branches, originating at 2/3 of wing length; single crossvein between RP and M; M approximated to R basally, diverging close to origin of RP; M with two branches, originating just distally to 1/2 of wing length; occurrence of arculus and up to

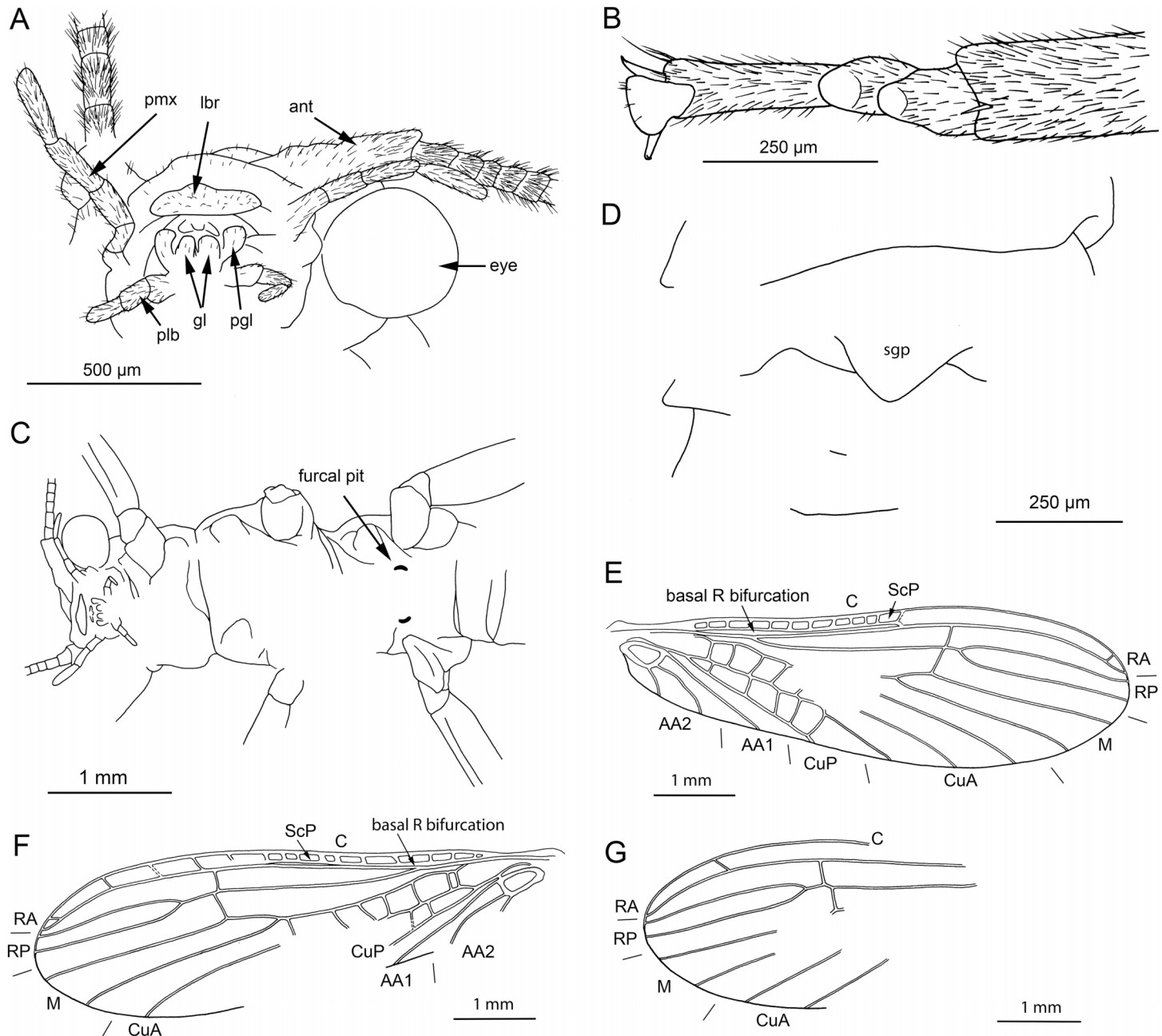

**Figure 2 *Petroperla mickjaggeri* gen. nov. sp. nov., holotype SMNS BU-79, line drawings.** (A) Head in frontoventral view. Abbreviations: ant, antenna; gl, glossa; lbr, labrum; pgl, paraglossa; pmx, maxillary palp; plb, labial palp. (B) Tarsus in ventral view. (C) Head and thorax in ventral view. (D) Subgenital plate; sgp. (E) Right forewing. (F) Left forewing. (G) Left hind wing.

six additional crossveins between M and CuA; CuA branched distally, with three branches; up to five crossveins between CuA and CuP; CuP simple and straight; AA1 simple; AA2 with two branches.

Hind wings (Fig. 2G, venation visible only in apical portion): length 5.6 mm; RA simple, almost reaching wing apex; single crossvein between RA and RP; RP with two distal

branches, originating at 2/3 of wing length; single crossvein between RP and M; M with two branches; branching of Cu and AA not recognizable.

Thoracic sterna (Fig. 2C) with one pair of dark patches centrally on mesosternum and two pairs on metasternum. Metasternum (Fig. 2C) with apparent arched furcal pits (not apparent on pro- and mesosternum due to state of preservation). Thoracic gill remnants absent.

Legs slender, covered with very short spine-like setae (thicker setae near posterior margin of femora and on dorsal surface of tibiae. Single tibial spur present apically (Fig. 2B). Tarsi with first two tarsomeres short (approximately equal in size) and apical tarsomere long, approximately 1.5× longer than first two combined (Fig. 2B). Arolium present, approximately as wide as width of apical tarsomere, with setae on arolium. Euplantulae present on tarsomeres 1 and 2.

*Abdomen.* Colour greyish, original colouration not preserved. Individual segments not distinctly enlarged posteromedially. No abdominal gill remnants recognizable. Subgenital plate poorly visible, probably triangular, without notch medially (Figs. 1D and 2D). Cerci short, 1 mm in length, covered with very short setae (Fig. 1A).

*Lapisperla* **gen. nov.**

urn:lsid:zoobank.org:act:05FAF13D-5548-4EFB-AFA1-CA30EA1A0B7B

**Type species.** *Lapisperla keithrichardsi*, gen. et sp. nov.

**Diagnosis.** By monotypy, as for the type species.

**Etymology.** The first part of the name refers to the Rolling Stones and is derived from Latin 'lapis,' meaning 'stone,' the suffix 'perla' refers to the stonefly genus *Perla*.

*Lapisperla keithrichardsi* **sp. nov.** (Figs. 3 and 4)

urn:lsid:zoobank.org:act:586DF169-34E5-4E6F-89EC-58E7D67BF6CB

**Diagnosis.** Prominent subgenital plate, abdominal segments extended posterolaterally, thoracic gill remnants present.

**Etymology.** The name refers to Keith Richards, founding member and guitar player of the Rolling Stones, master of the ancient art of weaving.

**Material.** Holotype specimen: SMNS BU-313, female.

**Description.**
Body length 6.1 mm (without head).

*Head.* Not preserved.

*Thorax.* Prothorax approximately quadrangular (Figs. 3A and 3B). Original cuticular colouration not preserved.

Forewings (Figs. 4A and 4B): right forewing present only as short basal fragment, 2.3 mm long. Left forewing almost complete, except for apical portion, length of preserved part approximately 6 mm, width 2.4 mm; costal field with nine visible crossveins, including area distal to ScP with two visible crossveins; ScP reaching RA just proximally to RA–RP crossvein; RA simple; single crossvein between RA and RP; RP originating from R

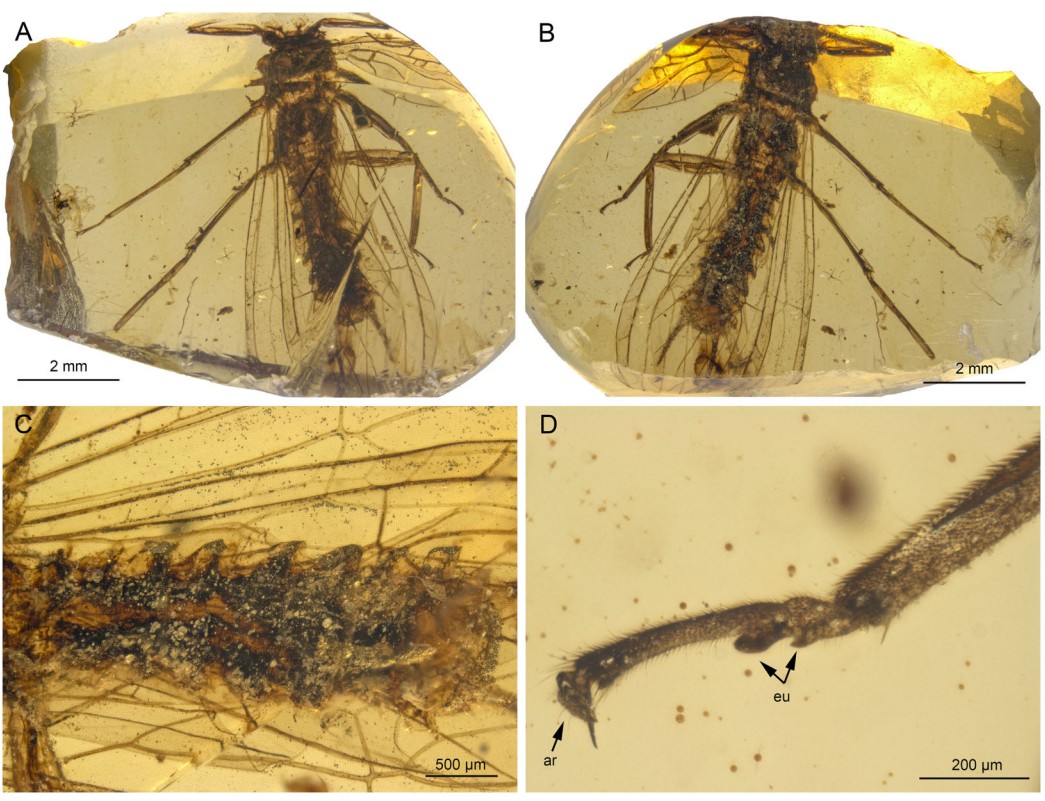

**Figure 3** *Lapisperla keithrichardsi* **gen. nov. sp. nov., holotype SMNS BU-313, photographs.**
(A) Dorsal view. (B) Ventral view. (C) Abdomen in ventral view. (D) Tarsus of right middle leg in lateral
view. Abbreviations: ar, arolium; eu, euplantula.

relatively basally, approximately 1.8 mm from wing base; RP with two visible distal
branches, originating just distally to RA–RP crossvein; single crossvein between RP and M;
M approximated to R basally; M with two visible branches, originating approximately
3.6 mm from wing base; occurrence of arculus and five additional crossveins between
M and CuA; CuA branched distally, with three branches; six crossveins between CuA
and CuP; CuP simple and straight; AA1 simple; AA2 with three branches.

Hind wings (Figs. 4C–4E): apical portions missing, length of preserved parts
approximately 5.7 mm (right hind wing) and 4.9 (left hind wing); costal field with six
visible crossveins, including area distal to ScP with three crossveins; ScP reaching RA
just proximally to RA–RP crossvein; RA simple, close to apex not approximated to C;
single crossvein between RA and RP; RP with two branches, originating just distally to
RA–RP crossvein; single crossvein between RP and M; M with two branches; CuA, CuP,
and AA1 simple. Venation pattern of AA2 not recognizable.

Visible gill remnants ventrolateral between meso- and metathorax. Legs slender,
covered with very short spine-like setae. Single apical tibial spur present (Fig. 3D).
Tarsi with first two tarsomeres short (approximately equal in size) and apical tarsomere
long, approximately 1.6× longer than first two combined (Fig. 3D). Long hair-like setae
near base of claws. Arolium present, approximately as wide as width of apical tarsomere.

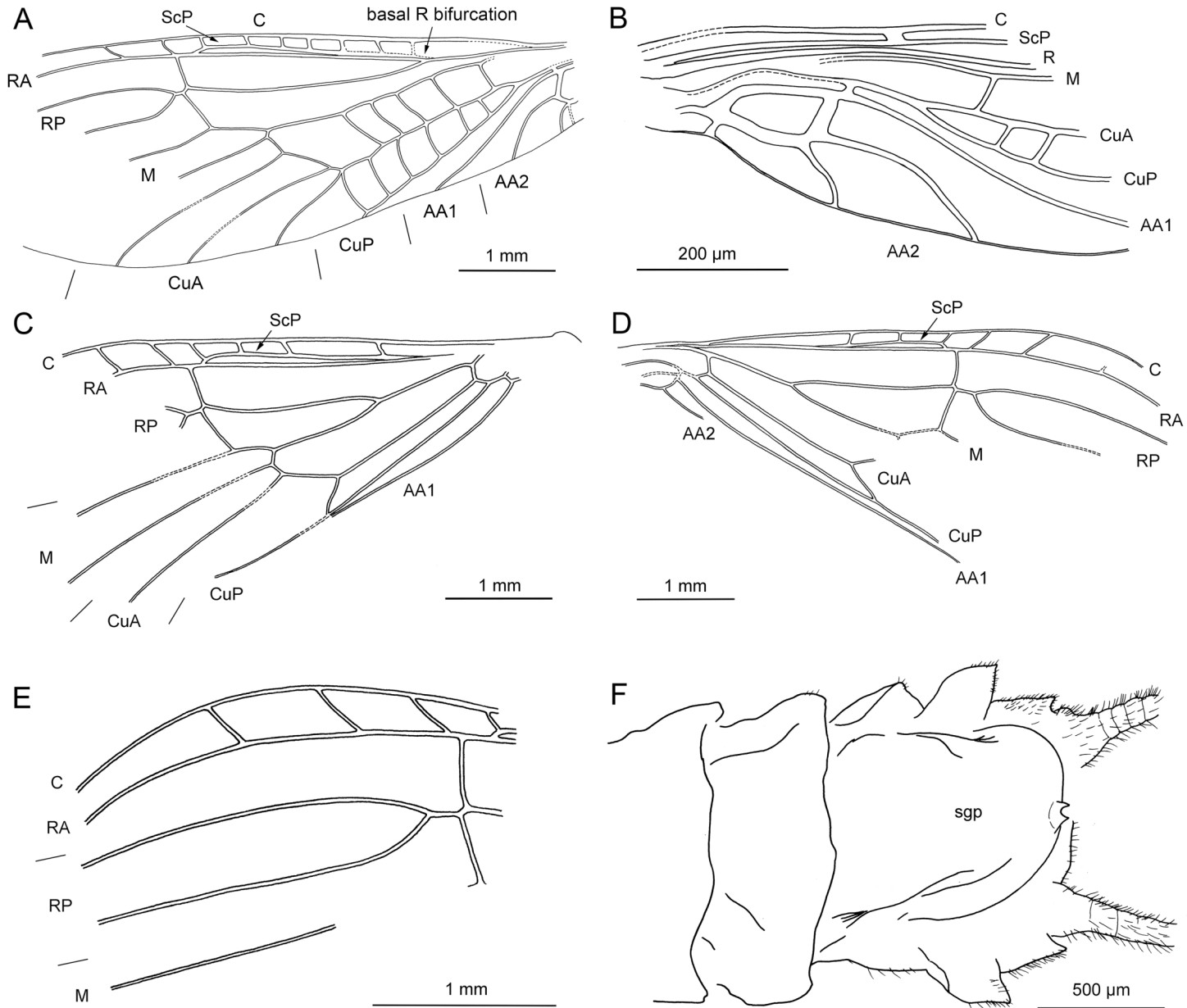

**Figure 4** *Lapisperla keithrichardsi* **gen. nov. sp. nov., holotype SMNS BU-313, line drawings.** (A) Left forewing. (B) Right forewing base. (C) Left hind wing. (D) Right hind wing. (E) Right hind wing from ventral side. (F) Tip of ventral abdomen with subgenital plate, sgp.

Arolium equipped with setae. Euplantulae present apically on tarsomeres 1 and 2 (Fig. 3D).

*Abdomen.* Individual segments with pronounced posterolateral extensions (Figs. 3C and 4F). No abdominal gill remnants recognizable. Prominent subgenital plate, rounded posteriorly, with slightly elevated notch posteromedially (Figs. 3C and 4F). Cerci broken off, only short basal parts preserved, densely covered with short setae.

**Remarks on generic composition of Petroperlidae.** *Lapisperla keithrichardsi* sp. nov. differs from *Petroperla mickjaggeri* sp. nov. in having a more prominent subgenital plate (badly visible, but much smaller in *Petroperla mickjaggeri* sp. nov.). Abdominal segments of *Lapisperla keithrichardsi* sp. nov. are equipped with posterolateral projections (no projections in *Petroperla mickjaggeri* sp. nov.), and meso- and metathorax of *Lapisperla keithrichardsi* sp. nov. ventrolaterally with gill remnants (apparently absent in *Petroperla mickjaggeri* sp. nov.). We consider these differences pronounced enough to justify a placement of these species in two separate genera within Petroperlidae.

Order Plecoptera Burmeister, 1839

Suborder Arctoperlaria Zwick, 1973

Infraorder Systellognatha Zwick, 2000

Superfamily Perloidea Latreille, 1802

Family Perlidae Latreille, 1802

Subfamily Acroneuriinae Klapálek, 1914

*Electroneuria*, **gen. nov.**

urn:lsid:zoobank.org:act:3A4CA23C-2D23-4F8E-9492-93C490AE736C

**Type species.** *Electroneuria ronwoodi*, gen. et sp. nov.

**Diagnosis.** By monotypy, as for the type species.

**Etymology.** Latin 'electrum' refers to both amber and electric guitars, the suffix '–neuria' to the stonefly subfamily Acroneuriinae.

*Electroneuria ronwoodi* **sp. nov.** (Figs. 5 and 6)

urn:lsid:zoobank.org:act:A52885BB-E2CF-49E8-871B-C4CE8577002C

**Diagnosis.** Larva with occipital spinule row complete medially; fringe of long thin setae laterally on pronotum; long hair-like setae on surface of wingpads and abdominal terga; posterior margin of abdominal terga with numerous very long thin setae; cerci long, with only short setae.

**Etymology.** The name of this immature specimen refers to Ronnie Wood, guitar player of the Rolling Stones since 1975, and youngest member of the Rolling Stones.

**Material.** Holotype specimen SMNS BU-306, larva.

**Description.** Body length 8.1 mm (Figs. 5A and 5B).

*Head.* Antennae 5.5 mm long (approximately 0.7× body length). Length of antennal segments in the middle third of antenna approximately equal to segment width. Mouthparts of predaceous type. Mandibles with only apical part of right mandible recognizable, with four rounded teeth. Left maxilla well visible (Figs. 5C and 6F), with thin galea slightly shorter than lacinia. Lacinia with two long, prominent pointed teeth, apical tooth longer. Row of six long setae on inner margin of lacinia, situated basally from subapical tooth, on slightly elevated hump. Left maxillary palp with three recognizable elongated palpomeres, basal part not well visible. Distal palpomere probably missing.
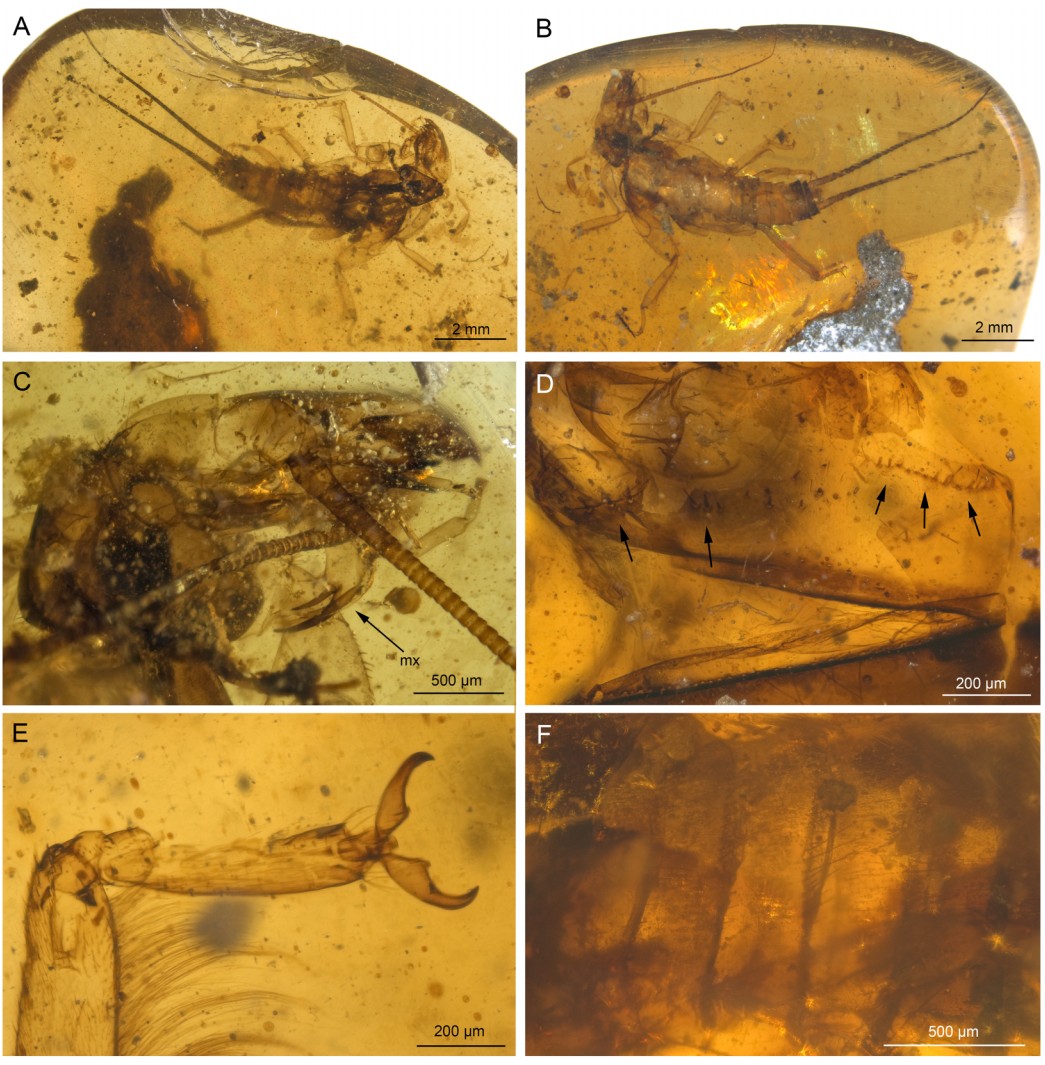

**Figure 5** *Electroneuria ronwoodi* **gen. nov. sp. nov., holotype SMNS BU-306, photographs.** (A) Dorsal view. (B) Ventral view. (C) Head with exposed maxilla, mx. (D) Head, occipital row of spines, marked by arrows. (E) Tarsus of left foreleg. (F) Setae on abdominal terga.

Sparse short setae present on all visible palpomeres, more dense on proximal palpomere. Other mouthparts not recognizable.

Occipital row of short spinules medially regular, in lateral parts scattered goups of longer spinules present (Figs. 5D and 6E).

*Thorax.* Pronotum covered with minute, hair-like setae on surface. Longer setae along margins (Fig. 6C). Pronounced wingpads (same size on meso- and metathorax), posterior notal contour not apparent. Thoracic gills badly preserved, indistinctly visible on left side between meso- and metathorax. Legs covered with numerous setae of various size and shape (Fig. 6G). Regular row of long, hair-like setae along outer margin of femora (length approximately 0.5× femur width) and tibiae (length approximately 1.5× tibia width). Tarsi with first two tarsomeres very short, third one approximately 2× longer
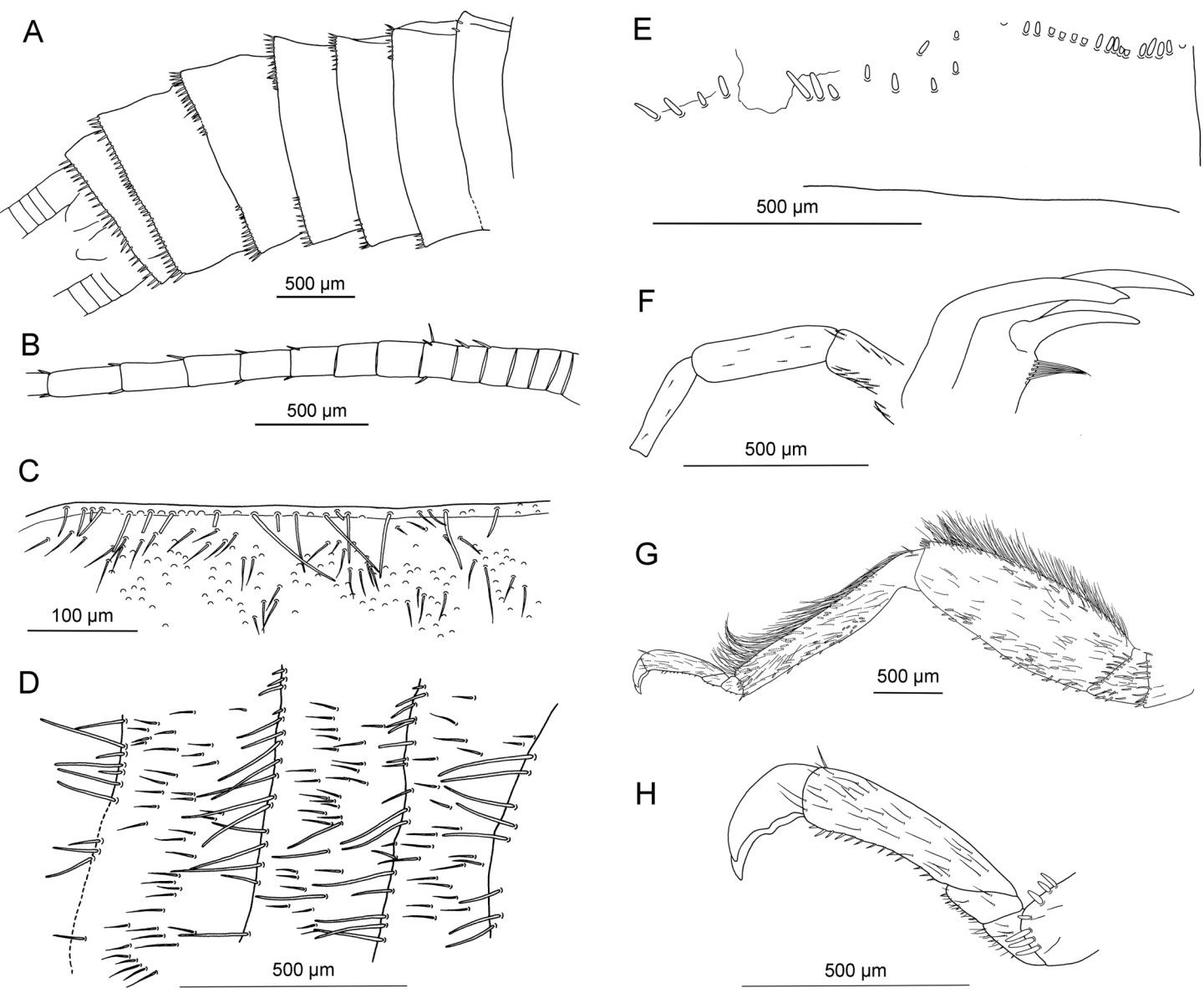

**Figure 6 *Electroneuria ronwoodi* gen. nov. sp. nov., holotype SMNS BU-306, line drawings.** (A) Abdominal sterna. (B) Detail of right cercus. (C) Anterior margin of pronotum. (D) Setation of abdominal terga. (E) Head, occipital row of spines. (F) Maxilla. (G) Right middle leg. (H) Tarsus of right middle leg.

than first two combined. Two claws with indistinctly pronounced denticles (Figs. 5E and 6H).

*Abdomen.* Posterior margin of terga with long setae (posterior tergal spinule fringe sensu *Stark & Gaufin, 1976*). Numerous long, intercalary setae on surface of terga (Figs. 5F and 6D). Sterna with short spine-like setae along posterior margin; row of these setae complete in two last segments (Fig. 6A). No abdominal gill remnants recognizable. Paraprocts bluntly pointed apically. Cerci 6 mm long (approximately 0.7× body length). Individual segments with short, spine-like setae (Fig. 6B).
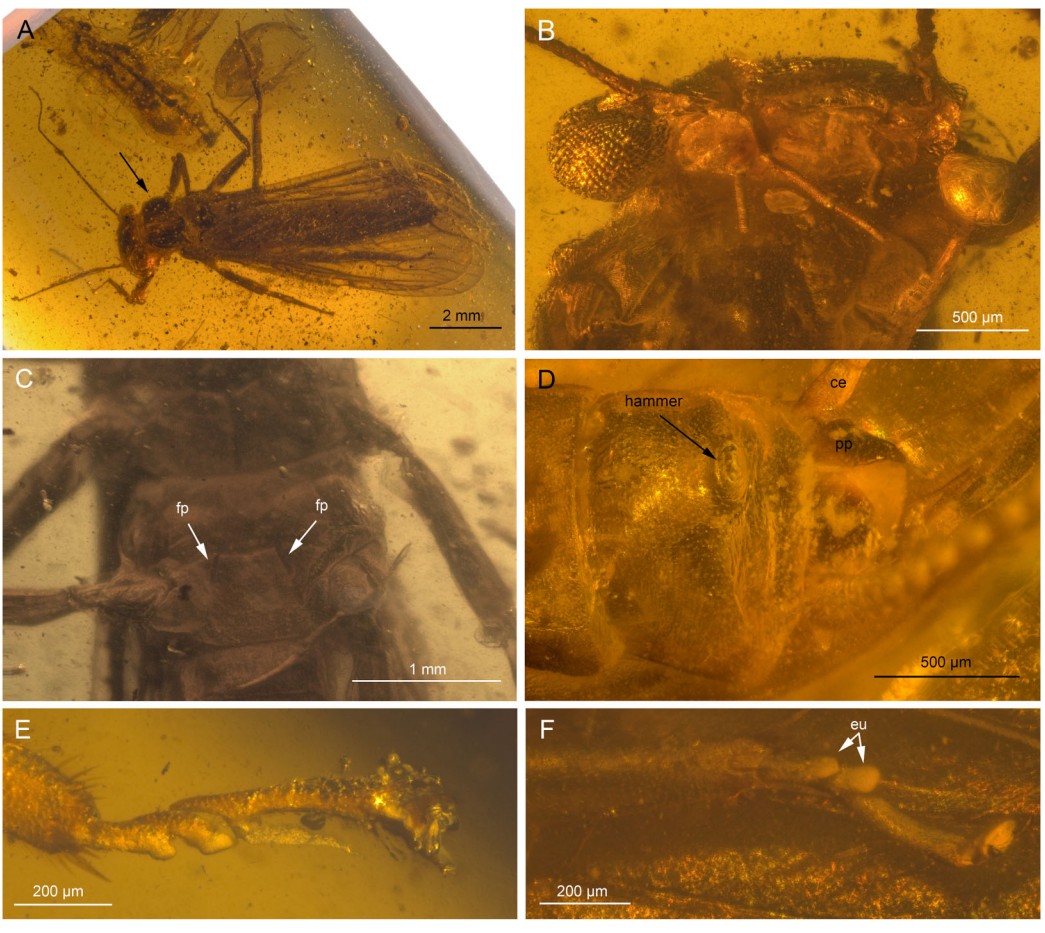

**Figure 7** *Largusoperla charliewattsi* **sp. nov., holotype SMNS BU-10, photographs.** (A) Dorsal view, arrow marks rounded pronotum. (B) Head in ventral view. (C) Meso- and metathorax in ventral view with furcal pits, fp. (D) Ventral tip of abdomen with hammer, paraprocts, pp, and cercus, ce. (E) Tarsus of right hind leg. (F) Tarsus of left hind leg with euplantulae, eu.

Order Plecoptera Burmeister, 1839

Suborder Arctoperlaria Zwick, 1973

Infraorder Systellognatha Zwick, 2000

Superfamily Perloidea Latreille, 1802

Family Perlidae Latreille, 1802

Subfamily Acroneuriinae Klapálek, 1914

Genus *Largusoperla* Chen et al., 2018

***Largusoperla charliewattsi* sp. nov.** (Figs. 7 and 8)

urn:lsid:zoobank.org:act:43BA0BA8-7818-47FE-BA37-E38F140A457C

**Diagnosis.** Large paraprocts; hammer knob-shaped, elongated transversally; pronotum not distinctly widened anteriorly.

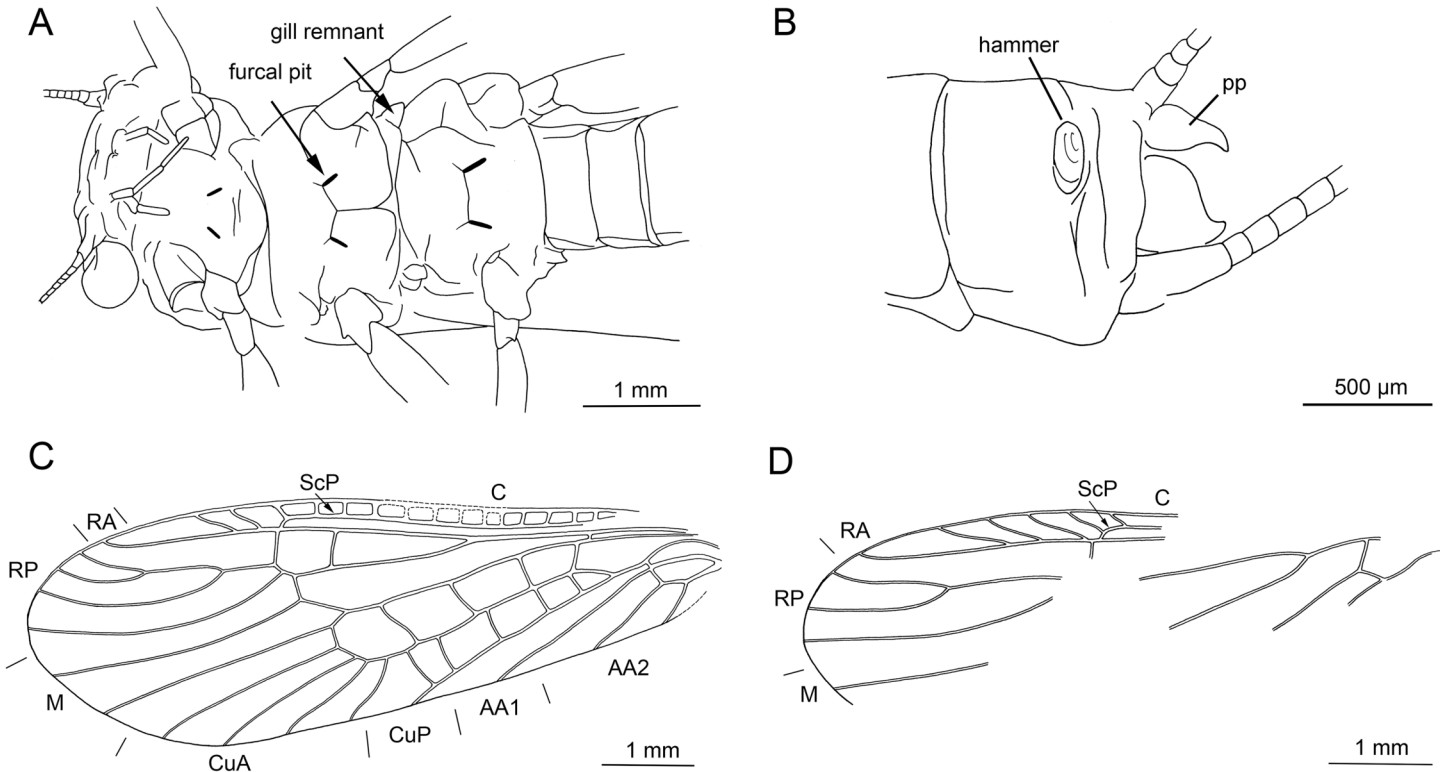

**Figure 8** *Largusoperla charliewattsi* **sp. nov., holotype SMNS BU-10, line drawings.** (A) Head and thorax in ventral view with furcal pits and gill remnants. (B) Ventral tip of abdomen with hammer and paraprocts, pp. (C) Left forewing. (D) Left hind wing.

**Etymology.** The name refers to Charlie Watts, drummer of the Rolling Stones, which is most adequate in regard of the pronounced drumming apparatus of the new species.

**Material.** Holotype specimen: SMNS BU-10, male.

**Description.** Body length 8.1 mm (Fig. 7A).

*Head.* Colour brown, marginal areas paler. Three fully developed ocelli. Median ocellus slightly subequal in size to lateral ocelli. Antennae 5.4 mm long (approximately 0.7× body length). Individual segments covered with short hair-like setae. Antennal segments in the middle third of antenna length approximately 2.5× longer than wide.

Maxillary palps slender, five elongated palpomeres covered with short hair-like setae (Fig. 7B). Labial palps rather elongated, three palpomeres present. First palpomere short (Fig. 7B). Other mouthparts not recognizable.

*Thorax.* Prothorax approximately quadrangular. Colour brown with pale median longitudinal line.

Forewings (Fig. 8C): length 8 mm, width 2.6 mm; costal field with 15 visible crossveins (including two crossveins distal to ScP); ScP reaching RA just proximally to 2/3 of wing length; RA simple; two crossveins between RA and RP; RP originating from R just distally to 1/3 of wing length; RP with four distal branches; single crossvein between RP and M; M slightly approximated to R basally; M with two branches, originating

just distally to 1/2 of wing length; occurrence of arculus and four additional crossveins between M and CuA; CuA branched distally, with four branches; five crossveins between CuA and CuP; CuP simple and straight; single cubito-anal crossvein reaching anal cell; A1 simple; AA2 with two branches, originating directly from anal cell.

Hind wings (Fig. 8D, venation details observable only in apical portion): length 6.9 mm; costal field with five visible crossveins (including three crossveins distal to ScP); ScP reaching RA distal to 1/2 of wing length; RA simple; single crossvein between RA and RP; RP originating from common stem with M at 1/4 of wing length; RP with three distal branches; branching of M, Cu and AA not recognizable.

Thoracic sterna (Figs. 7C and 8A) with apparent oblique furcal pits, converging anteriorly. Transverse meso- and metasternal ridge connecting anterior corners of respective furcal pits. Longitudinal median ridge apparent on mesofurcasternum. Y-ridge connecting posterior corners of furcal pits absent. Gill remnants recognizable on all thoracic segments (Fig. 8A).

Legs slender, covered with very short hair-like setae. Two thickened tibial spurs present apically on tibia, along with several additional long spine-like setae (Fig. 7E). Tarsi with first two tarsomeres short (approximately equal in size) and apical tarsomere long, approximately 1.5× longer than first two combined. Arolium present, slightly wider than width of apical tarsomere. Euplantulae present on tarsomeres 2 and 3 (Figs. 7E and 7F).

*Abdomen.* Colour brown, ventral side paler. Individual segments not distinctly enlarged posteromedially. Elevated, knob-shaped and transversally elongated hammer near posterior margin of sternum IX (Figs. 7D and 8B). Enlarged dark brown hook-like paraprocts between cerci (Figs. 7D and 8B). Cerci short, 2.1 mm in length (approximately 0.3× body length). Slightly moniliform in shape; segments covered with short hair-like setae (Fig. 7D).

**Affinities.** *Largusoperla charliewattsi* sp. nov. exhibits a combination of morphological characters allowing attribution to the family Perlidae (presence of long and slender palps, vestiges of thoracic gills, euplantulae, short first tarsomere, and forewings with numerous crossveins in the basal half of the costal field, see *Zwick, 1980*, *2000*). The presence of sclerotized and strongly recurved paraprocts and well-developed hammer allow a placement into the subfamily Acroneuriinae (*Stark & Gaufin, 1976*; *Zwick, 1980*, *2000*).

Within Acroneuriinae, *Largusoperla charliewattsi* sp. nov. can be attributed to the recently described genus *Largusoperla*, as defined by *Chen, Wang & Du (2018)*, based on the following diagnostic characters: triocellate; pronotum with pale median area; sternum IX with a small elevated lobe; large paraprocts sclerotized and strongly upcurved; short cerci, no longer than 1/2 of the abdomen length. Some characters stated by *Chen, Wang & Du (2018)* as diagnostic for the genus are not visible in *Largusoperla charliewattsi* sp. nov., possibly due to the state of preservation (head with dark stigma covering ocelli; abdominal terga with darker lateral markings). Another diagnostic character (abdominal segments posterolaterally extended) is not very pronounced in *Largusoperla charliewattsi* sp. nov. and is probably variable within the genus.

In *Pinguisoperla* Chen, 2018, the second Perlidae genus described from Burmese amber, the subfamilial attribution is uncertain (*Chen, 2018b*), and it might also belong to Acroneuriinae. Nevertheless, any affinity of *Largusoperla charliewattsi* sp. nov. to *Pinguisoperla* can be excluded based on the absence of enlarged, plump basal segment of the cercus, which constitutes the main diagnostic character of *Pinguisoperla*.

Up to now, four species of *Largusoperla* have been described, all based on male specimens from the Burmese amber, namely *Largusoperla acus* Chen et al., 2018, *Largusoperla flata* Chen et al., 2018, *Largusoperla arcus* Chen et al., 2018, and *Largusoperla difformitatem* Chen, 2018.

*Largusoperla charliewattsi* sp. nov. differs from *Largusoperla acus* in having less prominent posterolateral extensions on abdominal segments, pronotum not distinctly widened anteriorly, and paraprocts without needle-like apices. *Largusoperla flata* can be discriminated from *Largusoperla charliewattsi* sp. nov. by having entirely different shape of paraprocts (see *Chen, Wang & Du, 2018*, figs. 12 and 13; Figs. 7D and 8B). *Largusoperla arcus* possesses a more circular hammer on sternum IX, compared to a transversally rather elongated hammer in *Largusoperla charliewattsi* sp. nov. and paraprocts with apices distinctly diverging, contrary to more parallel oriented paraprocts in *Largusoperla charliewattsi* sp. nov. *Largusoperla difformitatem* differs by longer, highly divergent paraprocts and smaller hammer.

***Largusoperla billwymani* sp. nov.** (Figs. 9 and 10)

urn:lsid:zoobank.org:act:4AB77845-FAB5-4440-8496-935CBD1249B0

**Diagnosis.** Very large paraprocts, knob-shaped, circular hammer, slender body, trapezoidal pronotum.

**Etymology.** The name refers to Bill Wyman, former bass player of the Rolling Stones until 1991.

**Material.** Holotype specimen: SMNS BU-229, male.

**Description.** Body length 10.2 mm (Fig. 9A).

*Head.* Colour brown without distinct markings. Surface of head capsule with numerous short hair-like setae. Three fully developed ocelli. Median ocellus subequal in size to lateral ocelli. Antennae 7.6 mm long (approximately $0.7\times$ body length). Individual segments covered with short hair-like setae. Antennal segments in the middle third of antenna length approximately $2.5\times$ longer than wide.

Maxillary palps slender, with five elongated palpomeres covered with short hair-like setae (Figs. 9B and 10E). Palpomeres 1 and 5 short, palpomeres 2–4 approximately equal in length. Labium with glossae much smaller than paraglossae. Labial palps elongated, palpomere 2 longest, palpomeres 1 and 3 slightly shorter (Figs. 9B, 10C and 10E). All palpomeres covered with short hair-like setae. Other mouthparts not recognizable. *Thorax.* Prothorax trapezoidal, distinctly wider anteriorly. Surface of pronotum with numerous short hair-like setae. Colour brown with pale longitudinal median band, poorly visible due to state of preservation.

Forewings (Figs. 10A and 10B): length 9.3 mm, width 2.5 mm; costal field with up to 13 crossveins (single crossvein distal to ScP faintly visible on right forewing); ScP reaching

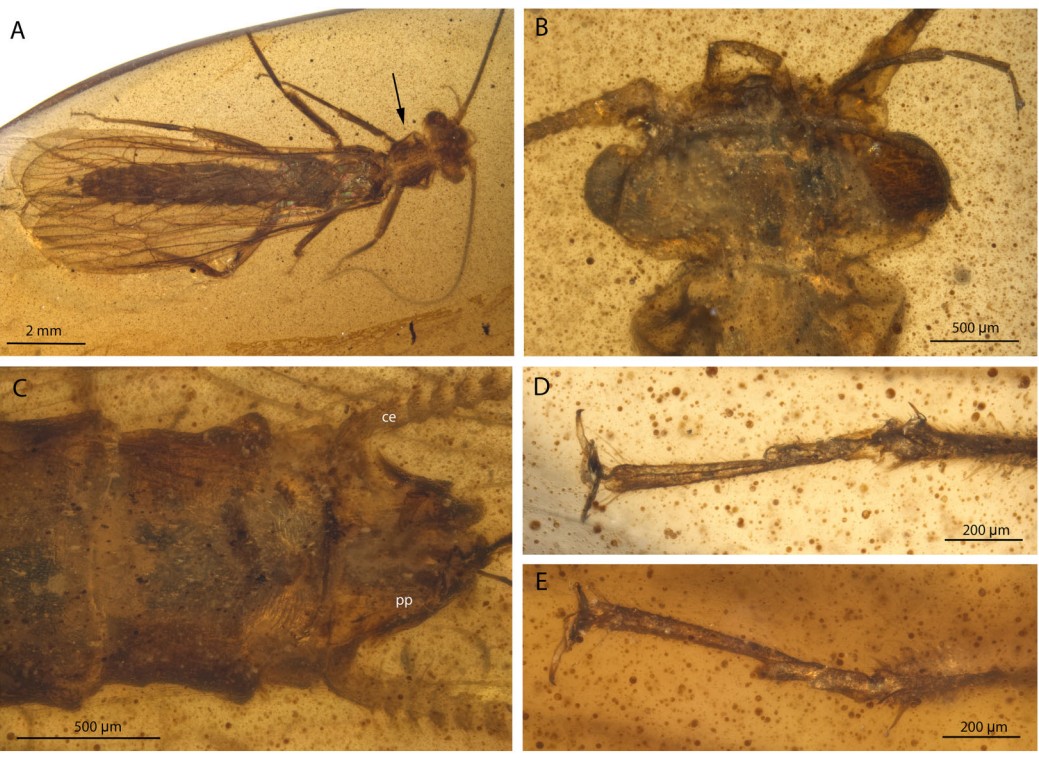

**Figure 9** *Largusoperla billwymani* **sp. nov., holotype SMNS BU-229, photographs.** (A) Dorsal view, arrow marks trapezoid pronotum. (B) Head in ventral view. (C) Ventral tip of abdomen with hammer, paraprocts, pp, and cercus, ce. (D) Tarsus of left hind leg. (E) Tarsus of right hind leg.

RA approximately at 2/3 of wing length; RA simple; single crossvein between RA and RP; RP originating from R distal to 1/3 of wing length; RP with four distal branches; single crossvein between RP and M; M approximated to R basally; M with two branches, originating proximally to 2/3 of wing length; occurrence of arculus and five additional crossveins between M and CuA; CuA branched distally, with three branches; five to six crossveins between CuA and CuP; CuP simple and straight; single cubito-anal crossvein reaching anal cell; AA1 simple; AA2 with two branches originating directly from anal cell.

Hind wings (Fig. 10D): length 7.8 mm; costal field with five visible crossveins (including single crossvein distal to ScP); ScP reaching RA proximal to 2/3 of wing length; RA simple; single crossvein between RA and RP; RP originating from common stem with M at 1/3 of wing length; RP with four distal branches; single crossvein between RP and M; branching of M, Cu and AA not recognizable.

Thoracic sterna (Fig. 10C) with apparent oblique furcal pits, converging anteriorly. Transverse meso- and metasternal ridge connecting anterior corners of respective furcal pits. Longitudinal median ridge apparent on mesofurcasternum, bifurcated and connecting anterior corners of furcal pits. Y-ridge connecting posterior corners of furcal pits absent. Gill remnants recognizable on all thoracic segments (Fig. 10C).

Legs slender, covered with short hair-like setae. Several longer and thicker spine-like setae occasionally on tibia. Two thickened tibial spurs and several setae apically on tibia

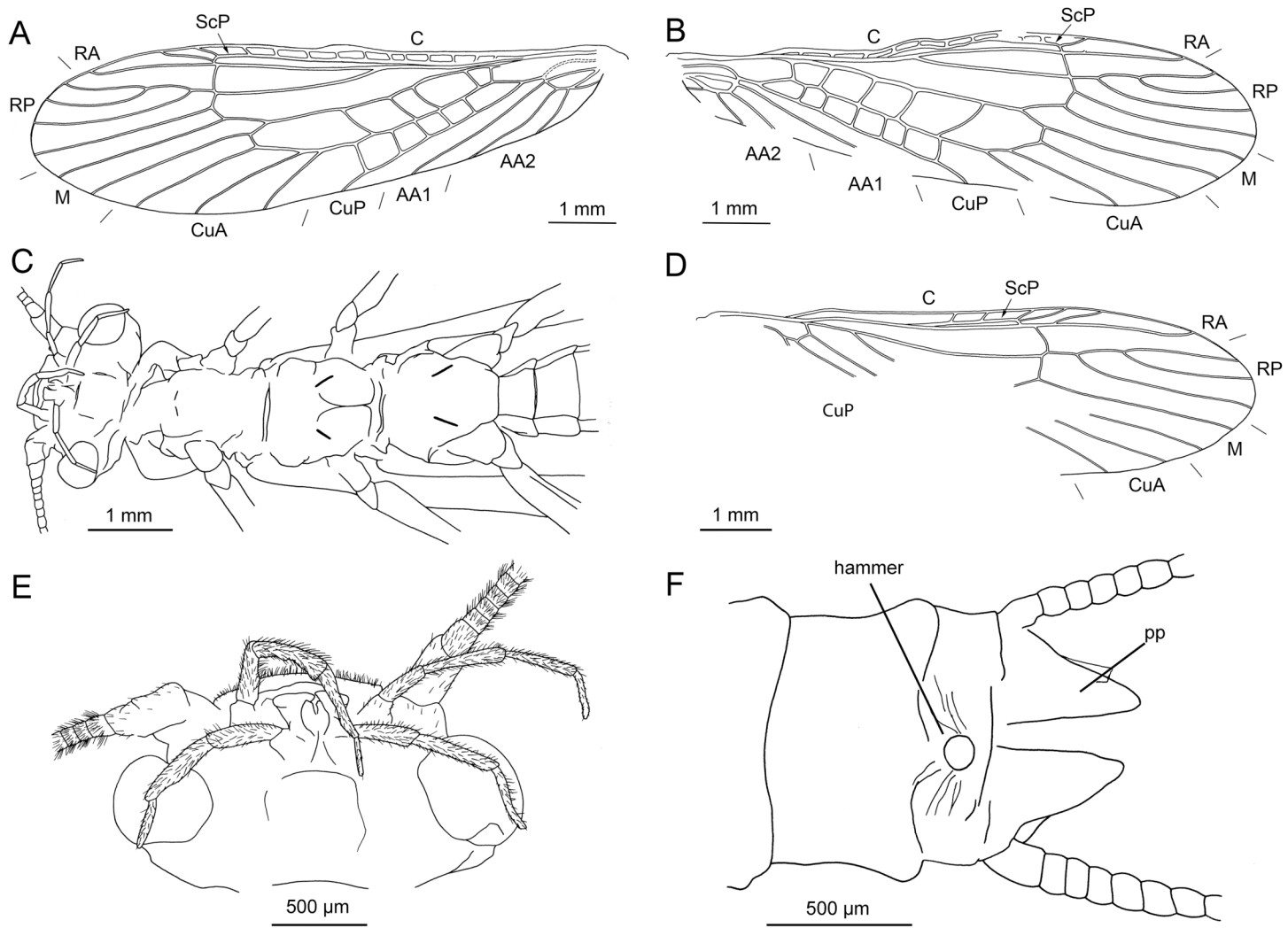

**Figure 10 _Largusoperla billwymani_ sp. nov., holotype SMNS BU-229, line drawings.** (A) Left forewing. (B) Right forewing. (C) Head and thorax in ventral view. (D) Right hind wing. (E) Head in ventral view. (F) Ventral tip of abdomen with hammer and paraprocts, pp.

(Figs. 9D and 9E). Very long hair-like setae subapically on tarsomere 3, approximately as long as claw length. Tarsi with first two tarsomeres short (approximately equal in size) and apical tarsomere long, approximately 1.6× longer than first two combined. Arolium present, slightly wider than width of apical tarsomere. Euplantulae present on tarsomeres 2 and 3 (Fig. 9E).

_Abdomen._ Colour brown, ventral side paler. Individual segments not distinctly enlarged posteromedially. Elevated, knob-shaped and circular hammer near posterior margin of sternum IX (Figs. 9C and 10F).

Enlarged hook-like paraprocts between cerci (Figs. 9C and 10F). Paraprocts with convex lateral margins, not diverging apically. Number of hair-like setae apparent on sterna VIII, IX (Fig. 9C).

Cerci very short, consisting of 13 segments, 1.7 mm in length (approximately 0.2× body length). Slightly moniliform in shape; segments covered with long hair-like setae (Fig. 9C).

**Affinities.** *Largusoperla billwymani* sp. nov. share all the diagnostic characters of the Perlidae subfamily Acroneuriinae and the genus *Largusoperla*, as discussed above.

*Largusoperla billwymani* sp. nov. clearly differs from *Largusoperla charliewattsi* sp. nov. based on the general shape of body and wings (more slender with narrower wings in *Largusoperla billwymani* sp. nov.) and pronotum shape (distinctly trapezoidal in *Largusoperla billwymani* sp. nov., more oblong-shaped in *Largusoperla charliewattsi* sp. nov.). *Largusoperla billwymani* sp. nov. also exhibits larger paraprocts in comparison with *L. charliewattsi* sp. nov. and a different shape of hammer (larger, circular in *Largusoperla billwymani* sp. nov. contrary to transversally elongated in *Largusoperla charliewattsi* sp. nov.).

The paraprocts of *Largusoperla billwymani* sp. nov. are similar to *Largusoperla flata*, but slightly differ in shape, lateral contour being concave in *Largusoperla flata* (*Chen, Wang & Du, 2018*, figs. 12 and 13) and convex in *Largusoperla billwymani* sp. nov. (Figs. 9C and 10F). *Largusoperla billwymani* sp. nov. also exhibits narrower pronotum and more circular hammer compared to *Largusoperla flata*. Another difference can be found in the forewing venation, vein RP in *Largusoperla flata* is three-branched with bifurcation on the posterior branch (*Chen, Wang & Du, 2018*, fig. 11), whereas *Largusoperla billwymani* sp. nov. has a four-branched RP with bifurcations on anterior branches (Figs. 10A and 10B). Nevertheless, wing venation characters must be taken cautiously, since they are subject to large intraspecific variability (*Béthoux et al., 2011*).

Another two species *Largusoperla acus* and *Largusoperla arcus* have differently shaped paraprocts compared to *Largusoperla billwymani* sp. nov. (*Largusoperla acus* with strongly constricted, needle-shaped apices and *Largusoperla arcus* with apices distinctly curved outward). *Largusoperla difformitatem* can be distinguished from *Largusoperla billwymani* sp. nov. by having a smaller, transversally elongated hammer and also highly divergent paraprocts.

*Largusoperla micktaylori* **sp. nov.** (Figs. 11–14)

urn:lsid:zoobank.org:act:17E6ED82-03C8-4824-9176-51C34EFDF66F

**Diagnosis.** Posterior margin of subgenital plate with three long, narrow, apically pointed processes.

**Etymology.** The name refers to Mick Taylor, guitar player of the Rolling Stones between 1969 and 1975 with unmatched virtuosity and dexterity, which is reflected by the finger-like, three-lobed subgenital plate of the new species.

**Material.** Holotype specimen SMNS BU-227, female; paratype specimen SMNS BU-312, female.

**Description.** Body length 8.6–10.5 mm (Figs. 11A, 11B, 13A and 13B).

*Head.* Colour brown without distinct markings. Three fully developed ocelli. Median ocellus slightly subequal in size to lateral ocelli. Antennae 7.2–7.4 mm long (approximately

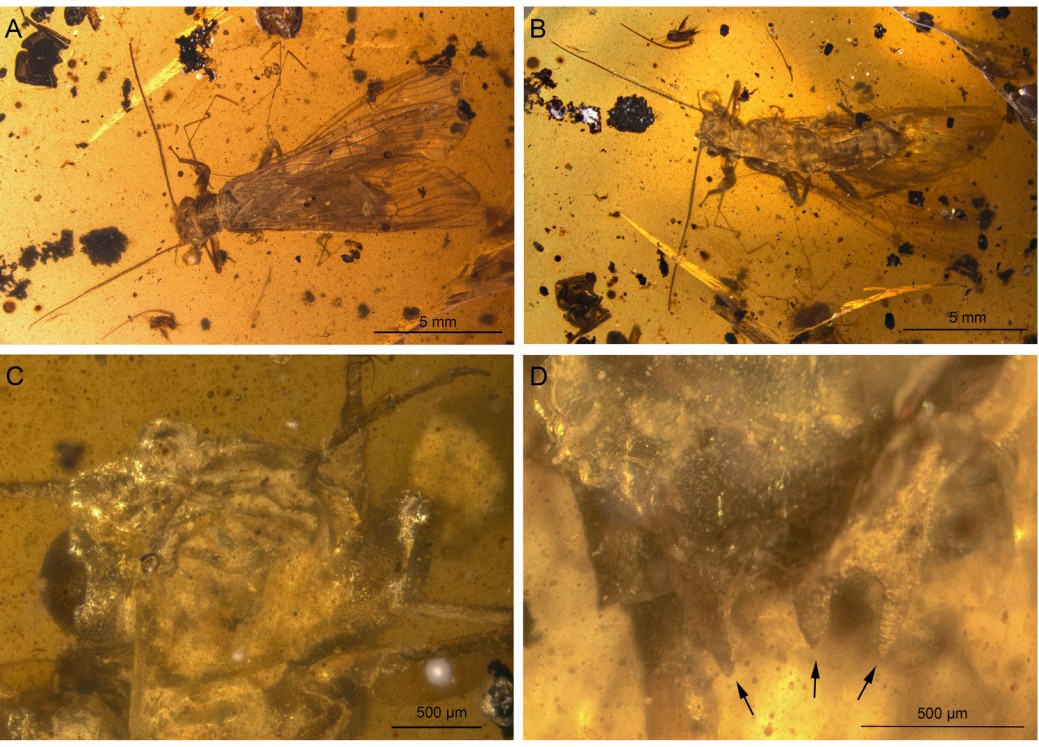

**Figure 11 *Largusoperla micktaylori* sp. nov., holotype SMNS BU-227, photographs.** (A) Dorsal view. (B) Ventral view. (C) Head in ventral view. (D) Three-lobed (arrows) subgenital plate.

0.8× body length). Individual segments covered with short hair-like setae. Antennal segments in the middle third of antenna length approximately 2.2× longer than wide.

Maxillary palps slender, with five elongated palpomeres covered with short hair-like setae (Figs. 11C and 13C). Palpomeres 1 and 5 short, palpomeres 2–4 approximately equal in length. Labium with glossae much smaller than paraglossae. Labial palps elongated, three-segmented. All palpomeres of approximately same length (Figs. 11C and 13C). All palpomeres covered with short hair-like setae. Other mouthparts not recognizable.

*Thorax.* Pronotum trapezoidal, anteriorly slightly wider. Colour brown with central longitudinal pale band.

Forewings (Figs. 12B, 12C, 14B and 14C): length 10–10.7 mm, width 3.1–3.5 mm; costal field with 9–14 crossveins, including one to two crossveins distal to ScP; ScP reaching RA just proximally to 2/3 of wing length; RA simple; single crossvein between RA and RP; RP originating from R just distally to 1/3 of wing length; RP with three to four distal branches; single crossvein between RP and M; M slightly approximated to R basally; M with two branches, originating proximal to 2/3 of wing length; occurrence of arculus and three to five crossveins between M and CuA; CuA branched distally, with four branches; three to seven crossveins between CuA and CuP; CuP simple and straight; single cubito-anal crossvein reaching anal cell; AA1 simple; AA2 with three branches originating directly from anal cell.

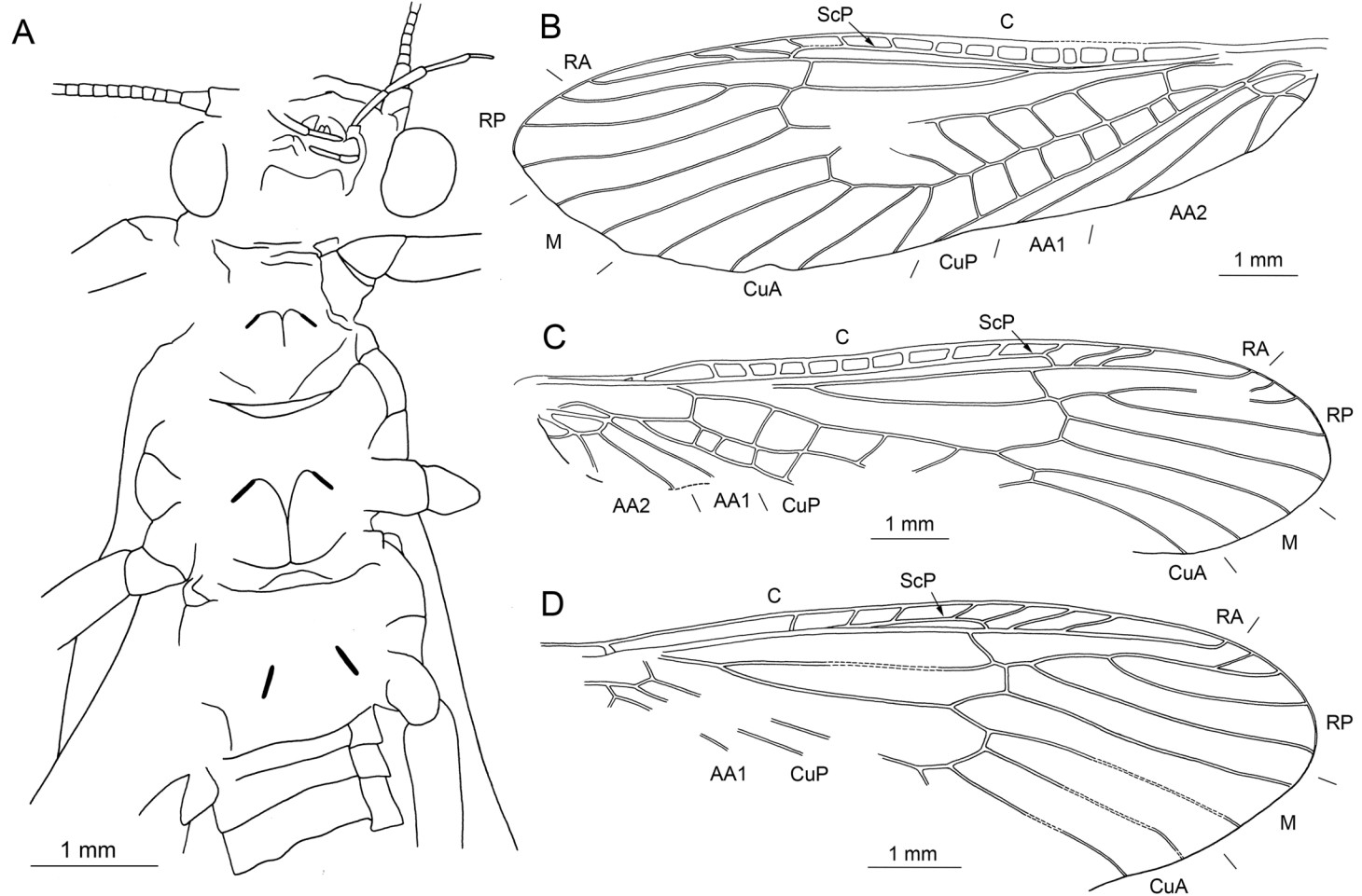

**Figure 12 *Largusoperla micktaylori* sp. nov., holotype SMNS BU-227, line drawings.** (A) Head and thorax in ventral view. (B) Left forewing. (C) Right forewing. (D) Right hind wing.

Hind wings (Figs. 12D, 14D and 14E): length 8.7–8.8 mm; costal field with seven to eight crossveins, including one to two crossveins distal to ScP; ScP reaching RA distal to 1/2 of wing length; RA simple; single crossvein between RA and RP; RP originating from common stem with M at approximately 1/4 of wing length; RP with three to four distal branches; single crossvein between RP and M; M with two branches originating distal to 1/2 of wing length; CuA with two branches; CuP and A recognizable only partially.

Thoracic sterna (Fig. 12A) with apparent oblique furcal pits, converging anteriorly. Longitudinal median ridge apparent on mesofurcasternum, bifurcated in the middle of its length and connecting anterior corners of furcal pits. Y-ridge connecting posterior corners of furcal pits absent. Transverse meso- and metasternal ridge connecting anterior corners of respective furcal pits indistinct. Remnants of thoracic gills recognizable.

Legs slender, covered with short hair-like setae. Two thickened tibial spurs apically on tibia. Very long hair-like setae subapically on tarsomere 3, approximately as long as claw length. Tarsi with first two tarsomeres short (approximately equal in size) and apical tarsomere long, approximately 1.5× longer than first two combined. Arolium present,

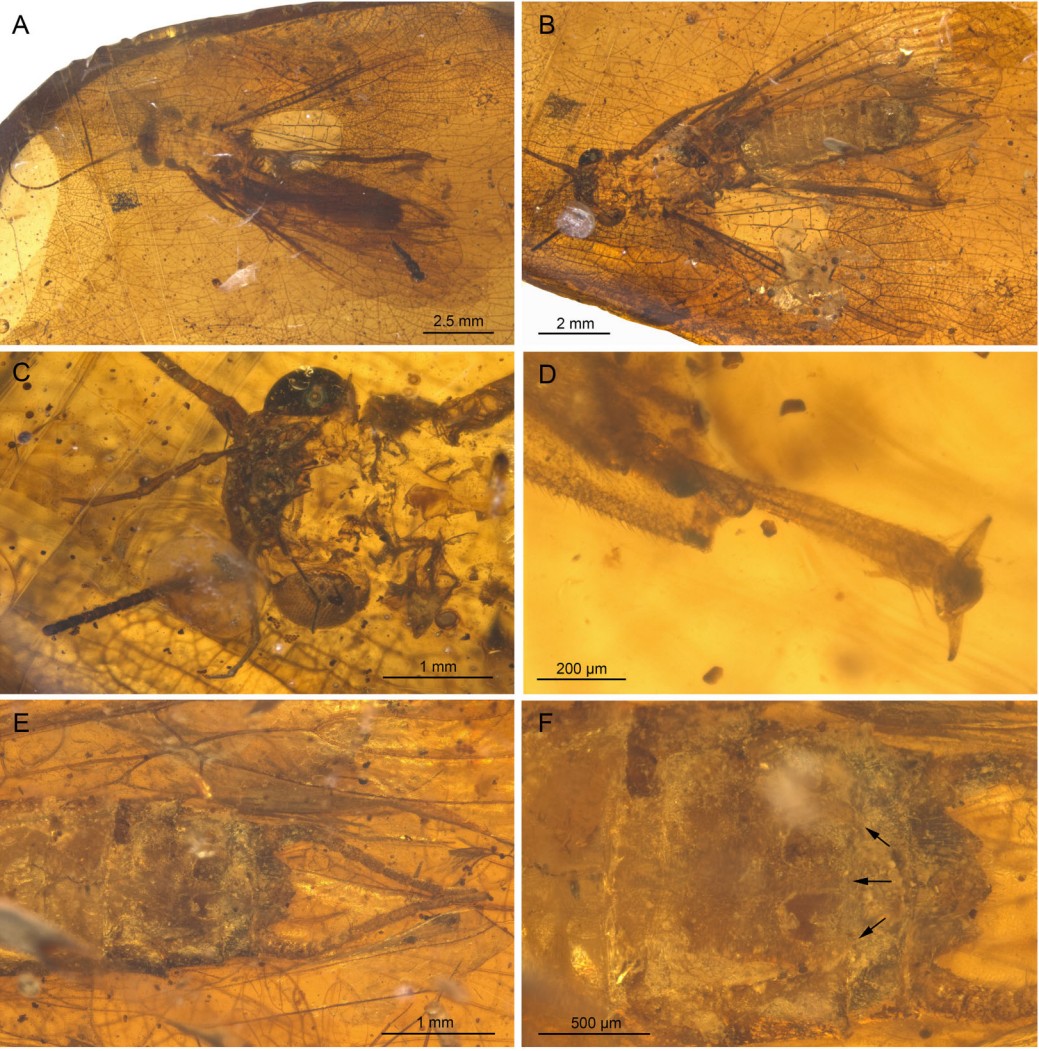

**Figure 13** *Largusoperla micktaylori* **sp. nov., paratype SMNS BU-312, photographs.** (A) Dorsal view. (B) Ventral view. (C) Head in ventral view. (D) Tarsus of right hind leg. (E) Ventral tip of abdomen with subgenital plate, paraprocts, and cerci. (F) Three-lobed (arrows) subgenital plate.

slightly wider than width of apical tarsomere. Euplantulae prominent, present on tarsomeres 2 and 3.

*Abdomen.* Individual segments not distinctly enlarged posterolaterally. Cerci short, 2.2–2.3 mm in length (approximately 0.2× body length), covered with short setae (Figs. 13E and 14A). Subgenital plate with three long, narrow, apically pointed processes on posterior margin (Figs. 11D, 13E, 13F and 14A). All three processes of same length, one situated medially and two laterally.

**Affinities.** *Largusoperla micktaylori* sp. nov. shares with male representatives of the genus *Largusoperla* traits typical for the Perlidae (presence of long and slender palps, vestiges of thoracic gills, euplantulae, short first tarsomere, and forewings with numerous crossveins in the basal half of the costal field). Relative size of glossae and paraglossae

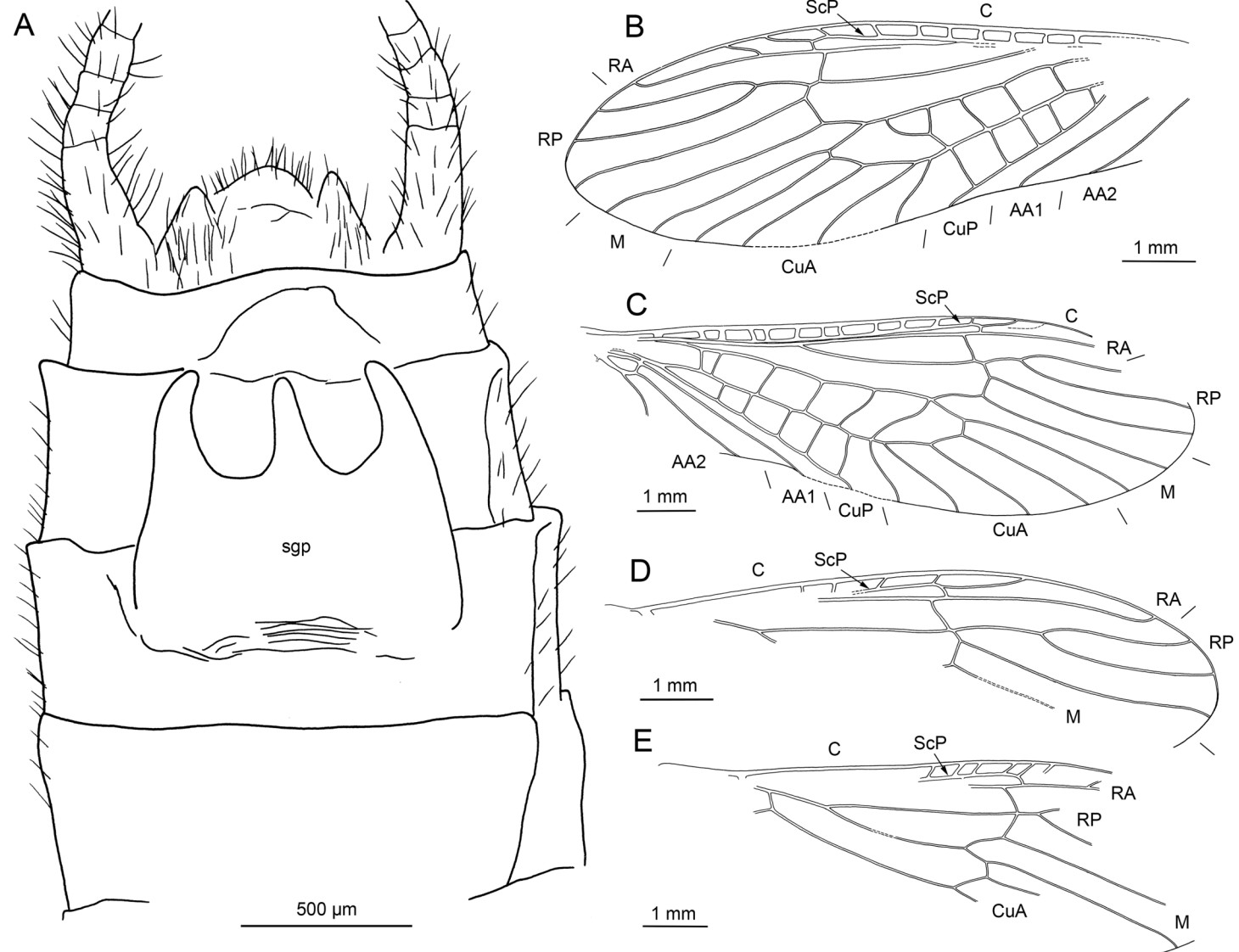

**Figure 14 *Largusoperla micktaylori* sp. nov., paratype SMNS BU-312, line drawings.** (A) Ventral tip of abdomen with subgenital plate, paraprocts, and cerci. (B) Left forewing. (C) Right forewing. (D) Left hind wing (ventral view). (E) Right hind wing.

is apparent, with paraglossae distinctly larger (Figs. 11C and 13C) and thus congruent with the assignment to Perlidae. The placement to the subfamily or tribe within Perlidae can be done only based on characters of male genitalia, which is inapplicable for *Largusoperla micktaylori* sp. nov. However, based on the presence of key generic diagnostic characters of *Largusoperla* (triocellate; pronotum with pale median area; short cerci, no longer than 1/2 of the abdomen length) together with overall high similarity in the arrangement of tarsi, mouthparts, wing venation and simultaneous occurrence with *Largusoperla* males, we include *Largusoperla micktaylori* sp. nov. into *Largusoperla*. From *Pinguisoperla* Chen, 2018, another Perlidae genus described from the Burmese amber, *Largusoperla micktaylori*

sp. nov. clearly differs by the absence of an enlarged, plump basal segment of the cercus, which constitutes the main diagnostic character of *Pinguisoperla*.

The subgenital plate in *Largusoperla micktaylori* sp. nov. with three pronounced extensions on posterior margin is rather unusual and unique among known stoneflies and represents a crucial diagnostic character of *Largusoperla micktaylori* sp. nov. Since the shape of the subgenital plate is exactly the same in both specimens and since they also share other morphological characteristics (body size, arrangement of mouthparts, general wing venation pattern), we consider these specimens to represent a single new species.

Individual specimens of *Largusoperla micktaylori* sp. nov. differ from each other in the wing venation details such as number of crossveins and RP branches (Figs. 12B, 12C, 14B and 14C). However, these differences correspond with common intraspecific variability, as documented for recent Perlidae by *Béthoux et al. (2011)*. In the holotype, the number of RP branches even differs between right and left forewing, being four and three, respectively, a phenomenon not uncommon in extant taxa.

### *Largusoperla brianjonesi* sp. nov. (Figs. 15 and 16)

urn:lsid:zoobank.org:act:3FF737E9-D935-48D7-901A-A1F5FBF61CE9

**Diagnosis.** Posterior margin of subgenital plate with two broadly rounded lobes.

**Etymology.** The name refers to Brian Jones, founding member and former guitar player of the Rolling Stones until 1969.

**Material.** Holotype specimen SMNS BU-311, female.

**Description.** Body length 10.3 mm (Figs. 15A and 15B).

*Head.* Colour brown, with distinct pale areas (Fig. 15C). Median ocellus slightly subequal in size to lateral ocelli. Antennae 6.4 mm long (approximately 0.6× body length). Individual segments covered with short hair-like setae. Antennal segments in middle third of antenna length approximately 2× longer than wide.

Maxillary palps with three visible elongated palpomeres of approximately same length. Two basal palpomeres not visible, presumably shorter than three distal palpomeres. Glossae and paraglossae not recognizable. Postmentum large. Labial palps rather elongated, three-segmented. First palpomere shortest, second slightly longer, apical palpomere longest (Fig. 16F).

*Thorax.* Pronotum slightly trapezoidal, wider anteriorly (Fig. 15A). Colour brown, pale longitudinal band medially.

Forewings (Figs. 16B and 16C): length 10.8 mm, width 3.1 mm; costal field with 14 crossveins visible (including single crossvein distal to ScP); ScP reaching RA approximately at 2/3 of wing length; RA simple; single crossvein between RA and RP; RP originating from R distal to 1/3 of wing length; RP with three distal branches; single crossvein between RP and M; M approximated to R basally; M with two branches, originating proximal to 2/3 of wing length; occurrence of arculus and six additional crossveins between M and CuA; CuA branched distally, with three branches; three

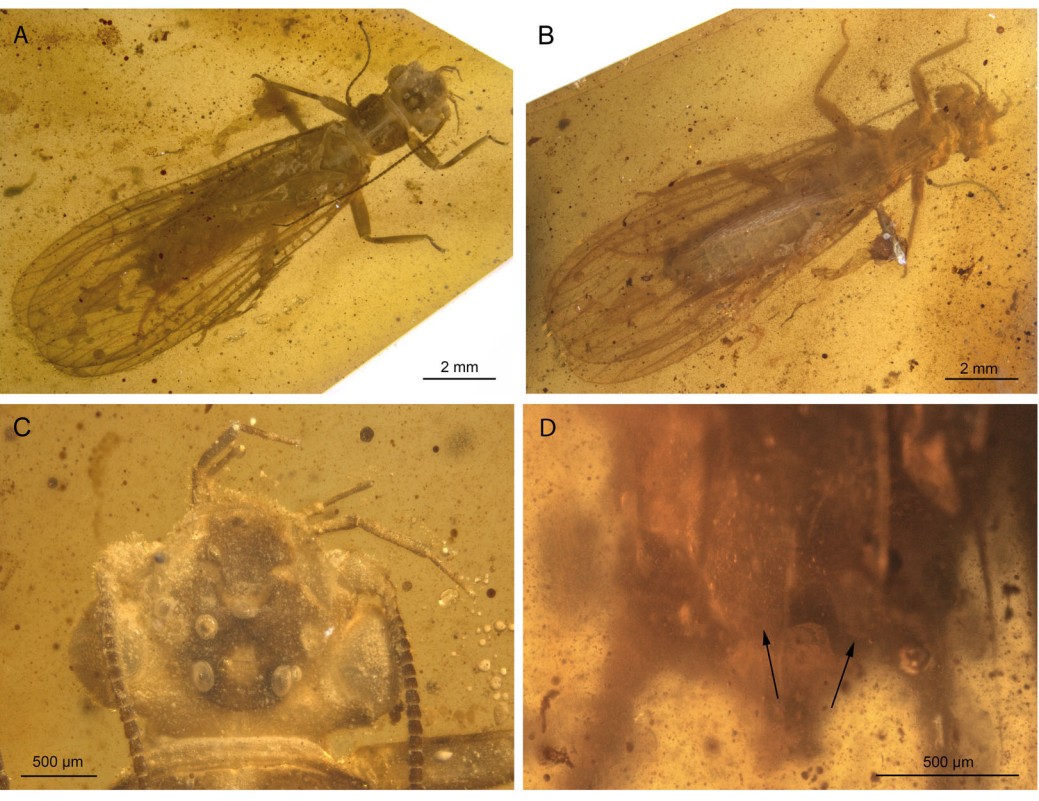

**Figure 15** *Largusoperla brianjonesi* **sp. nov., holotype SMNS BU-311, photographs.** (A) Dorsal view.
(B) Ventral view. (C) Head in dorsal view. (D) Bi-lobed (arrows) subgenital plate.

crossveins between CuA and CuP; CuP simple and straight; AA1 simple; AA2 with two
branches originating directly from anal cell.

Hind wings (Fig. 16D, details of venation only visible in apical portion): length 8.9 mm;
costal field with seven visible crossveins (including two crossveins distal to ScP); ScP
reaching RA distal to 1/2 of wing length; RA simple; single crossvein between RA and RP;
RP originating from common stem with M approximately at 1/4 of wing length;
RP with four distal branches; single crossvein between RP and M; course and branching
of M, CuP and AA unrecognizable.

Thoracic sterna (Fig. 16A) with apparent oblique furcal pits, converging anteriorly.
Transverse meso- and metasternal ridge connecting anterior corners of respective furcal
pits. Longitudinal median ridge apparent on mesofurcasternum, connecting anterior
corners of furcal pits. Y-ridge connecting posterior corners of furcal pits absent.
Remnants of thoracic gills recognizable on all thoracic segments.

Legs slender, covered with short hair-like setae. One thickened tibial spur and several
setae apically on tibia (Fig. 16E). Tarsi with first two tarsomeres short (approximately
equal in size) and apical tarsomere long, approximately 1.5× longer than first two
combined. Arolium present, slightly wider than width of apical tarsomere. Euplantulae
present on tarsomeres 2 and 3 (Fig. 16E).
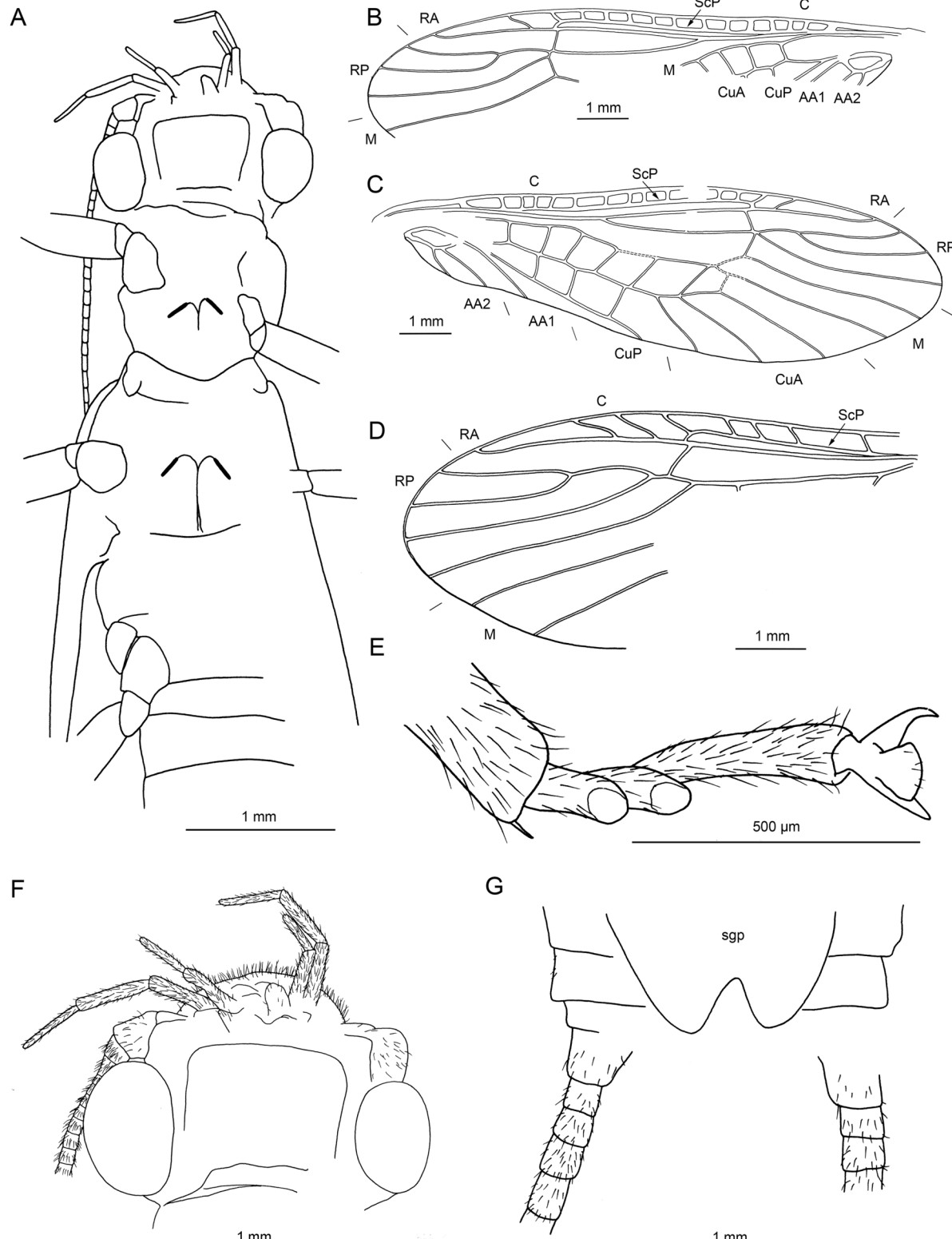

**Figure 16 *Largusoperla brianjonesi* sp. nov., holotype SMNS BU-311, line drawings.** (A) Head and thorax in ventral view. (B) Left forewing. (C) Right forewing. (D) Right hind wing (ventral view). (E) Tarsus in ventral view. (F) Head in ventral view. (G) Bi-lobed subgenital plate.

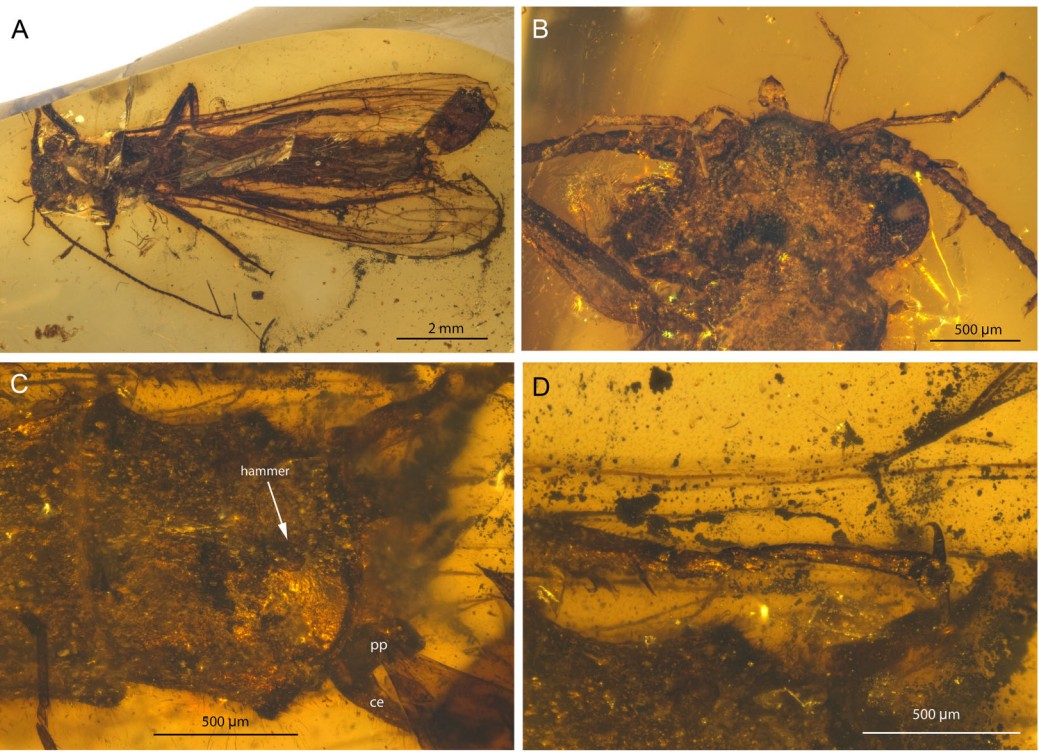

**Figure 17 *Largusoperla* sp., SMNS BU-228, photographs.** (A) Dorsal view. (B) Head in ventral view. (C) Ventral tip of abdomen with hammer, paraprocts, pp, and cercus, ce. (D) Tarsus of left hind leg.

*Abdomen.* Individual segments not distinctly enlarged posterolaterally. Cerci very short, 1.9 mm in length (approximately 0.2× body length). Subgenital plate bilobed with wide rounded notch in middle of posterior margin (Figs. 15D and 16G).

**Affinities.** *Largusoperla brianjonesi* sp. nov. is very similar to *Largusoperla micktaylori* sp. nov., sharing all the diagnostic characters for the family Perlidae and genus *Largusoperla* as discussed above. At the same time, *Largusoperla brianjonesi* sp. nov. clearly differs from *Largusoperla micktaylori* sp. nov. based on the shape of the subgenital plate (with two broad lobes on the posterior margin in *Largusoperla brianjonesi* sp. nov. in contrast to the presence of three narrow projections in *Largusoperla micktaylori* sp. nov.).

***Largusoperla* sp.** (Figs. 17 and 18)
**Material.** SMNS BU-228 (Figs. 17 and 18)
**Perlidae: Acroneuriinae sp.** (Fig. 19)
**Material.** SMNS BU-99 (Fig. 19).

**Remarks to species descriptions.** We refrain from associating males and females described in *Largusoperla*, since any association would be purely speculative. Clearly, multiple similar species lived in the common palaeohabitat.

The larva described above as *Electroneuria ronwoodi* sp. nov. is attributable to the subfamily Acroneuriinae, as all the adults described here in *Largusoperla* and possibly

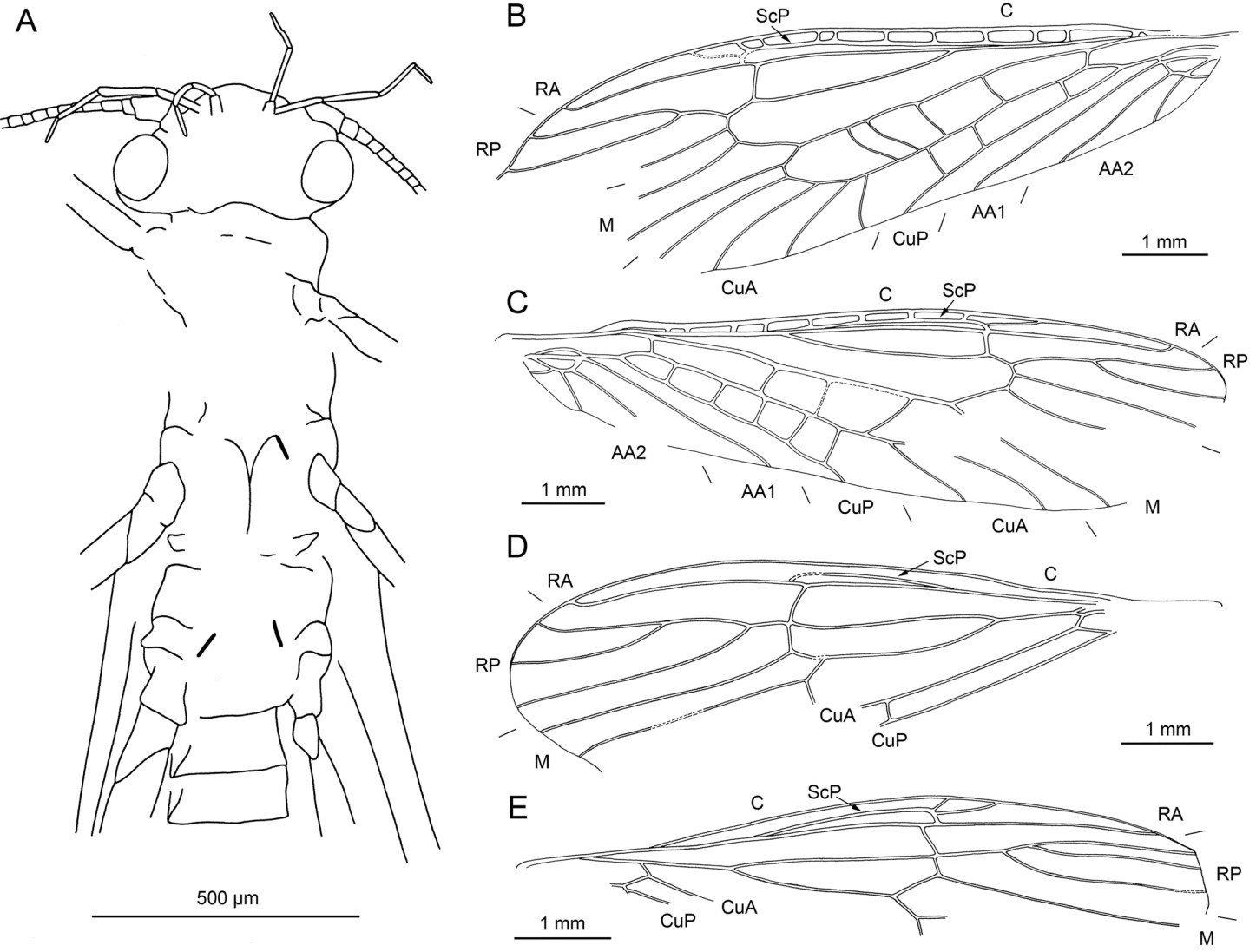

**Figure 18** *Largusoperla* **sp., SMNS BU-228, line drawings.** (A) Head and thorax in ventral view. (B) Left forewing. (C) Right forewing. (D) Left hind wing. (E) Right hind wing.

also adults of *Pinguisoperla*, described by *Chen (2018b)*. However, we refrain from assigning *Electroneuria ronwoodi* sp. nov. into one of these two genera, based on the difference in the length of cerci. Usually there is a correlation between the length of cerci in larvae and adults in Systellognatha (*Zwick, 1980*). *Electroneuria ronwoodi* sp. nov. exhibits much longer cerci than *Largusoperla* and *Pinguisoperla*, therefore we prefer assigning this species to a separate genus at this time. As *Electroneuria ronwoodi* has an age of about 100 million years, we also exclude its placement within an extant genus of Acroneuriinae.

Discriminating species characters for males of Acroneuriinae are the mainly detailed shape of hammer and paraprocts (*Stark, Froehlich & Zuñiga, 2009*). For females, the crucial structure is the shape of subgenital plate. Therefore, we designate species only for

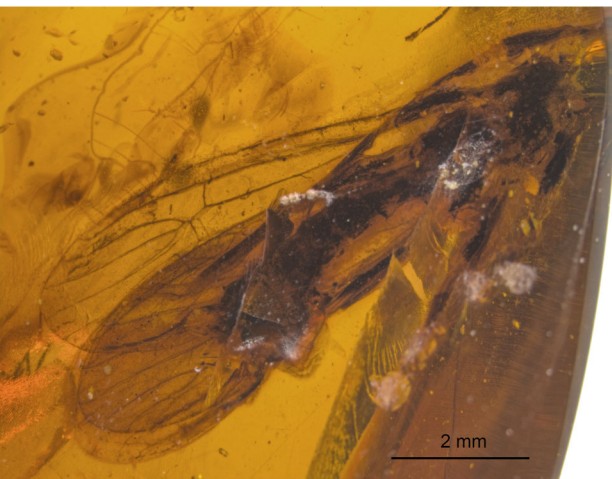

2 mm

**Figure 19 Acroneuriinae spp., SMNS BU-99, photograph.** Dorsal view.

the specimens with these structures well visible. We studied another male specimen (SMNS BU-228) attributable to *Largusoperla*, but with hammer and/or paraprocts insufficiently visible to allow detailed comparisons. Yet another investigated specimen (SMNS BU-99) is assignable to Acroneuriinae, but without visible genitalia and generally too poorly preserved to allow generic attribution.

## DISCUSSION

### Phylogenetic position of Petroperlidae

The absence of crossveins in the distal half of (ScP+) RA–RP area allows the placement of Petroperlidae in the suborder Arctoperlaria (*Cui et al., 2015*).

Among characters of phylogenetic significance, segmentation of tarsi is well visible in the specimens. The first two tarsomeres are short, approximately equal in size and the apical tarsomere is long, approximately 3.5× longer than first tarsomere (Figs. 2B and 3D). A short basal tarsomere is either considered an autapomorphy of Perloidea (*Zwick, 1973*), a synapomorphy of Perloidea with Peltoperlidae + Styloperlidae (*Zwick, 1980*), or as a character independently derived in these lineages (*Zwick, 2000*; *Nelson, 2009*). In any case, it points to the placement of Petroperlidae in the infraorder Systellognatha. Any assignment to the infraorder Euholognatha is excluded because of the presence of euplantulae on tarsomeres 1 and 2, absent in the Euholognatha (*Nelson, 2009*). Further diagnostic characters of Systellognatha are the apomorphic presence of setae on the arolium (see Figs. 2B and 3D) (*Nelson, 2009*) and the plesiomorphic presence of numerous crossveins in the costal field, both of which also accounts for *Petroperla* gen. nov. and *Lapisperla* gen. nov. (Figs. 2E and 4A).

Moreover, in the forewing of *Lapisperla keithrichardsi* sp. nov., the basal crossvein in the costal field (Fig. 4B) is stronger than the remaining crossveins, short, and of opposed obliquity (in *Petroperla mickjaggeri* sp. nov. the view onto this character is obstructed). This character has been proposed as diagnostic for Systellognatha by *Cui et al. (2015)*.

Within Systellognatha, a labium with glossae only slightly smaller than paraglossae and with rather stout labial palps (Figs. 1C and 2A) excludes a closer relationship of *Petroperla mickjaggeri* sp. nov. to Perloidea, the latter characterized by apomorphic shape of labium with glossae much shorter than paraglossae (*Zwick, 2000*). Therefore, based on the shape of labium, a relationship to the other systellognathan superfamily, Pteronarcyoidea, would seem more probable. However, neither *Petroperla* gen. nov. nor *Lapisperla* gen. nov. possess any of the wing venation characters known to represent traits of individual Pteronarcyoidea lineages (*Cui et al., 2016*), which are numerous crossveins between M and CuA in distal half of wing; AA2 with more than three branches; and M with more than two branches. Moreover, Petroperlidae feature several wing venation characters not present in Systellognatha. These include a vein RA almost reaching the wing apex, and a plesiomorphic origin of vein RP in the forewing close to wing base (Figs. 2E, 2F and 4A). Vein RA almost reaching the wing apex is absent in Carboniferous stem-stonefly *G. carpenteri* and is not known for any other stonefly taxa, both extant and fossil, except of the genus *Kargaloperla Sinitshenkova, 1987* from the Upper Permian of Ural (Palaeoperlidae, see *Sinitshenkova, 1987*, fig. 10). It is unlikely that it represents a plesiomorphic character that was independently lost in several other basal stonefly lineages (including *G. carpenteri*). It is more parsimonious to assume that it is either an autapomorphy of Petroperlidae, or even a synapomorphy of Petroperlidae and *Kargaloperla*.

An origin of RP close to the wing base presumably represents a plecopteran ground plan character known already from Paleozoic stoneflies. It is also present in *G. carpenteri*, and according to *Béthoux et al. (2011)* it is a diagnostic character of early stoneflies, also supporting the affinity of *G. carpenteri* to Plecoptera. It is also present in Permian *Palaeoperla exacta* Sharov, 1961 (Palaeoperlidae) and *Perlopsis filicornis* Martynov, 1940 (Perlopseidae) and can be observed in Euholognatha as well, e.g. recent Leuctridae or fossil Perlariopseidae (*Sinitshenkova, 1992*, fig. 2). Probably the origin of RP in the forewing situated close to the wing base is also retained in Petroperlidae and lost in all other Systellognatha.

Based on available evidence, we propose the phylogenetic position of Petroperlidae as a distinct stem lineage of Systellognatha. The presence of a short first tarsomere is evaluated here as potential apomorphic character of Systellognatha and was already early present in the stemline of Systellognatha. Further support for a placement in Systellognathan stemline is the apomorphic presence of an arolium with setae. Tarsal euplantulae and numerous crossveins in the costal field are plesiomorphies shared with remaining Systellognatha. The plesiomorphic proximal origin of RP excludes the position of Petroperlidae in the crown group Systellognatha, and RA reaching the wing apex might be an autapomorphic character of Petroperlidae. Vein RA almost reaching the wing apex is not completely preserved in *Lapisperla keithrichardsi* sp. nov. due to the missing wing apices. However, from the course of RA in right hind wing approximating the wing apex (Figs. 4D and 4E), we assume a similar pattern in *Petroperla mickjaggeri* sp. nov. Therefore, as *Lapisperla keithrichardsi* sp. nov. and *Petroperla mickjaggeri* sp. nov. share the same unique combination of characters, we assign them to the same family Petroperlidae.

Relationships with several fossil groups considered as stem-Systellognatha (*Illies, 1965*; *Stark & Gaufin, 1976*) cannot be reliably analysed. Permian Tshekardoperlidae Sinitshenkova, 1987 and Jurassic Platyperlidae Sinitshenkova, 1982 are known in the larval stage only (*Sinitshenkova, 1987*; *Carpenter, 1992*). The family Palaeoperlidae Sharov, 1961, whose members sometimes exhibit aforementioned wing venation characters, represent a very vaguely defined Permian group known only from larvae and isolated forewings and thus many other characters (tarsi, mouthparts) cannot be compared. Association of larvae and adults described as Palaeoperlidae is also uncertain. In *Sinitshenkova (2002)*, Palaeoperlidae was considered as a sister group of Perlina (containing Zwick's Systellognatha + Antarctoperlaria), in other words one of the most basal branches within Plecoptera. However, wing venation indicates at least some members of Palaeoperlidae (genus *Kargaloperla*) might be related to Petroperlidae, and Palaeoperlidae might not constitute a monophyletic group.

## Systematic position of *Electroneuria ronwoodi* gen. nov. sp. nov

Predaceous mouthparts with long and slender palps are diagnostic for larvae of the superfamily Perloidea within the arctoperlariid group Systellognatha (*Zwick, 1980*).

Of the three families of Perloidea (Perlidae, Perlodidae, and Chloroperlidae), the wing pad arrangement of *Electroneuria ronwoodi* sp. nov. rather resembles Perlidae and Chloroperlidae, since the 'rounded shape of larval wing-pads whose edges meet medially, without leaving a separate notal contour' is mentioned as a possible synapomorphy of these two families within Perloidea by *Zwick (2000)*.

*Electroneuria ronwoodi* sp. nov. can be excluded from Chloroperlidae, as larvae of this family exhibit a slender body with usually an oval pronotum and cerci distinctly shorter than the abdomen (see examples in *Brittain & Saltveit, 1996*; *Stewart & Stark, 2002*), presumably apomorphic features of the family (*Zwick, 2000*). On the contrary, the body of *Electroneuria ronwoodi* sp. nov. is relatively robust bearing long cerci. Another apomorphic character of Chloroperlidae (except for subfamily Paraperlinae) is a thin, asymmetrically inserted, terminal maxillary palpomere (*Zwick, 1973*, *1980*, *2000*; *Surdick, 1985*), which is not present in *Electroneuria ronwoodi* sp. nov.

Thus more likely is a placement within Perlidae, also supported by the presence of thoracic gills (*Zwick, 1980*). Thoracic gills are difficult to identify in *Electroneuria ronwoodi* sp. nov. due to its poor preservation. The specimen most probably is an exuvia, as only the exoskeleton is preserved. Furthermore, respective body regions corresponding with the insertion points of thoracic gills are obscured by mineral particles, but the remains of thoracic gills might be visible at least on the left side of the body, between meso- and metathorax. Its affinity to Perlidae is further confirmed by the presence of an occipital row of short spinules (*Stark & Gaufin, 1976*; *Sivec, Stark & Uchida, 1988*; *Zwick, 2000*). The arrangement of this row represents the larval key character distinguishing Perlidae subfamilies Perlinae and Acroneuriinae (*Stark & Gaufin, 1976*). In *Electroneuria ronwoodi* sp. nov., the row is slightly sinuate, regular medially, in lateral parts disintegrate into scattered goups of longer spinules (Figs. 5D and 6E), an arrangement present in

Acroneuriinae, in contrast to the straight and complete occipital row of Perlinae (*Stark & Gaufin, 1976*; *Sivec, Stark & Uchida, 1988*).

There are also two Permian families, Palaeoperlidae and Tshekardoperlidae, which might be candidates for a closer relationship to *Electroneuria* gen. nov., but both of them are rather poorly defined. Diagnoses are mostly based on the length ratios of body parts (*Sinitshenkova, 1987*). Both Palaeoperlidae and Tshekardoperlidae differ from *Electroneuria ronwoodi* (apart from the marked difference in geological layer) in the shape of thoracic segments with clearly visible notal contour between wingpads (figs. 8, 25–27 in *Sinitshenkova, 1987*). Furthermore, Tshekardoperlidae are characterized by prolonged antennal segments, 2–3 times longer than wide in the middle third of the antenna (*Sinitshenkova, 1987*). *Electroneuria ronwoodi* sp. nov. exhibits much shorter antennal segments (Fig. 5C). Jurassic Platyperlidae most notably differ by the presence of a median projection on the posterior margin of terminal abdominal segment (*Carpenter, 1992*; *Sinitshenkova, 1987*) which is missing in *Electroneuria ronwoodi* sp. nov.

Based on the combination of characters as presented above, we attribute *Electroneuria ronwoodi* to the Perlidae subfamily Acroneuriinae. There are no larval diagnostic characters for the three Acroneuriinae tribes (*Stark & Gaufin, 1976*; *Murányi & Li, 2016*), but from all described genera of Acroneuriini, Anacroneuriini, and Kiotinini, the larval specimen *Electroneuria ronwoodi* sp. nov. can be diagnosed by the combination of characters as follows: (1) occipital spinule row complete medially, (2) fringe of long thin setae laterally on pronotum (most of them broken, but bases clearly visible), (3) long hair-like setae on surface of wing pads and abdominal terga, (4) posterior margin of abdominal terga with numerous very long thin setae, and (5) long setae on cerci absent.

## Palaeodiversity, Palaeoecology, and Palaeobiogeography of Perlidae

The family Perlidae, despite being very abundant and diversified today, was so far only scarcely found in the fossil record. Several specimens of the extant genus *Perla* are known from Eocene Baltic amber (*Pictet & Hagen, 1856*; *Carpenter, 1992*). A single immature larva of *Perla* cf. *burmeisteriana* Claassen, 1936 was described by *Prokop (2002)* from Lower Miocene deposits of the Czech Republic. An adult specimen of *Dominiperla antigua Stark & Lenz, 1992* was described from Dominican amber (dated between upper Eocene to lower Miocene) and attributed to Acroneuriinae (subfamily 'Anacroneuriinae,' as stated in *Stark & Lenz, 1992* probably represents a spelling error). Affinities were evaluated based on the course of M1 vein, only slightly divergent from R until the origin of Rs (present in some Perlidae), and small, spindle-shaped eggs similar to the Neotropical genus *Anacroneuria*. A young larva is also known from Oligocene of SW Montana, attributed to ?*Acroneuria* (*Lewis & Gundersen, 1987*). Another larva was described as *Euperlida parvicercifera* Cifuentes-Ruiz, 2007 from Oligocene of Mexico (*Cifuentes-Ruiz et al., 2007*) and tentatively attributed to Perlidae subfamily Acroneuriinae based on general body resemblance (the shape and size of the head, wing pads, body pattern, foreleg, and pronotum shape).

The oldest Perlidae species *Archaeoperla rarissimus* Liu, Ren & Sinitshenkova, 2008 described by *Liu, Sinitshenkova & Ren (2008)* from Jurassic/Cretaceous of China, was

later synonymized by *Cui et al. (2015)* with *Sinosharaperla zhaoi* Liu et al., 2007, which probably represents stem-Systellognatha, although attribution to Perlidae was not excluded by *Cui et al. (2015)*. If we do not consider *S. zhaoi* as a genuine Perlidae, the genera *Largusoperla, Pinguisoperla,* and *Electroneuria* from the mid-Cretaceous represent the oldest record of the family Perlidae. At the same time, they represent fossils attributed to Perlidae without doubt and based on multiple characters. Previously published Perlidae records were assigned to this family mostly based on a limited number of characters, often only on a superficial resemblance.

Three tribes were proposed within Perlidae subfamily Acroneuriinae based on adult male characters (Acroneuriini, Anacroneuriini, and Kiotinini, see *Stark & Gaufin, 1976*; *Stark, Froehlich & Zuñiga, 2009*; *Murányi & Li, 2016*). Regarding the position of *Largusoperla*, we concur with *Chen, Wang & Du (2018)* that the form of the hammer, an elevated knob rather than a low callus, suggests affinity to Anacroneuriini, whereas the well-developed anterior ocellus and short cerci represent plesiomorphies within Perlidae and suggest a possible position of *Largusoperla* in the stemline of Acroneuriini + Anacroneuriini.

Out of 10 stonefly specimens described in the present study, eight are attributed to the subfamily Acroneuriinae and, apart from one larval specimen, belong to the single genus *Largusoperla*, as the four species described recently by *Chen, Wang & Du (2018)* and *Chen (2018c)*. A single specimen described by *Chen (2018b)* in the genus *Pinguisoperla* probably also belongs to the same subfamily. From such a pattern (even taking into consideration a fragmentary nature of the fossil record), we might assume that the stonefly community in the Burmese amber forest streams was dominated by representatives of a limited number of higher taxa. This might be caused by the nature of the original habitat, which has been assumed to be a tropical Araucaria forest (*Grimaldi, Engel & Nascimbene, 2002*). Present stonefly diversity is concentrated in the temperate regions, a pattern probably consistently followed throughout their history (*Sinitshenkova, 1997*). However, as demonstrated by our data, some groups like Acroneuriinae had been adapted to the warmer streams of the Cretaceous Burmese amber palaeohabitat. Although the adults might be speculated to represent allochthonous material in the inclusions, our finding of a larva of the same taxonomic placement rather corroborates the occurrence of Acroneuriinae in the Burmese amber forest streams.

With more than 1,000 described species, Perlidae is the most diverse extant family. The most diverse perlid areas are Asia and Central and South America. At present, the Chinese perlid species represent about 25% of the world Perlidae, making the Oriental region one of the most important areas of the world for perlid diversity (*Fochetti & De Figueroa, 2008*). We assume placement of all Acroneuriinae adult material dealt with in the present study into the tribe Anacroneuriini sensu *Stark & Gaufin (1976)*. At present, all extant Anacroneuriini are restricted to the Neotropics and southern portion of the Nearctic (*Murányi & Li, 2016*). *Stark & Gaufin (1976)* hypothesized recent Anacroneuriini arose from a now extinct Oriental group which dispersed across Northern Africa in Early Cretaceous and became established in South America. *Largusoperla* and *Electroneuria* gen. nov. might actually represent such a stem-Anacroneuriini group, still to be found in the Oriental region during the late Cretaceous and subsequently becoming extinct.

Close relations between Neotropic and Oriental fauna dating back to the Cretaceous were already known in other groups of aquatic and terrestrial insects, e.g. the Burmese amber damselfly family Mesomegaloprepidae, related to extant Neotropical taxa (*Huang et al., 2017*) or the psocid family Compsocidae, known from Burmese amber and the extant fauna of Central America (*Azar, Hakim & Huang, 2016*; *Sroka & Nel, 2017*). Our study thus provides support for the hypothesis of *Stark & Gaufin (1976)* claiming oriental origin of Anacroneuriini.

## CONCLUSIONS

Our findings report with the fossil family Petroperlidae the presence of a new taxon in the stem lineage of Systellognatha in the Cretaceous. It reveals new insights into the character evolution within this group: Petroperlidae have shortened tarsomeres 1 and 2, which consequently must now be assumed as ground plan character of Systellognatha. Otherwise, the Burmese Cretaceous stonefly fauna was dominated by species of the Anacroneuriini (Perlidae: Acroneuriinae), a taxon, which today is restricted to the Nearctic realm. This points to an oriental origin of Anacroneuriini, thus confirming the respective hypothesis of *Stark & Gaufin (1976)*.

## ACKNOWLEDGEMENTS

This publication is dedicated to the Rolling Stones on the occasion of Mick Jagger's 75th birthday on 26 July 2018. In this way AHS wants to thank the Rolling Stones for the soundtrack of his life and commemorate the attendance of his 20th Rolling Stones concert in Stuttgart, Germany on 30 June 2018. We are obliged to Milan Pallmann (SMNS) for his help with macro photography. We sincerely thank Patrick Müller, Käshofen, Germany for access to his collection and donation of material.

### Funding

The Society for the Promotion of the Natural History Museum Stuttgart funded the acquisition of several specimens described in this contribution. This research received support for Pavel Sroka from the SYNTHESYS Project http://www.synthesys.info/, which was financed by European Community Research Infrastructure Action under the FP7 'Capacities' Program and institutional support RVO: 60077344. This publication received funding from the CSU Libraries Open Access Research and Scholarship (OARS) Fund. The funders had no role in study design, data collection and analysis, decision to publish, or preparation of the manuscript.

### Grant Disclosures

The following grant information was disclosed by the authors:
Society for the Promotion of the Natural History Museum Stuttgart.
'Capacities' Program and institutional support RVO: 60077344.
CSU Libraries Open Access Research and Scholarship (OARS) Fund.

## Competing Interests

The authors declare that they have no competing interests.

## Author Contributions

- Pavel Sroka analysed the data, prepared figures and/or tables, authored or reviewed drafts of the paper, approved the final draft.
- Arnold H. Staniczek analysed the data, contributed reagents/materials/analysis tools, prepared figures and/or tables, authored or reviewed drafts of the paper, approved the final draft.
- Boris C. Kondratieff analysed the data, authored or reviewed drafts of the paper, approved the final draft.

## Data Availability

All holotypes and paratypes described in this publication are stored within the SMNS amber collection, Stuttgart, Germany.

http://col.smns-bw.org/db/datenbank.php.

## New Species Registration

The following information was supplied regarding the registration of a newly described species:
Publication LSID: urn:lsid:zoobank.org:pub:486E9A01-EF59-41D7-B001-4AD6D7FBB11F
Petroperlidae: urn:lsid:zoobank.org:act:3F9EB209-A5DD-49C9-A5D6-7952F05F94A4
Petroperla: urn:lsid:zoobank.org:act:68557BD3-F4E1-43BB-A10B-68D796971769
*P. mickjaggeri*: urn:lsid:zoobank.org:act:838EDFF8-BD85-4F83-88F6-87E38701A941
Lapisperla: urn:lsid:zoobank.org:act:05FAF13D-5548-4EFB-AFA1-CA30EA1A0B7B
*L. keithrichardsi*: urn:lsid:zoobank.org:act:586DF169-34E5-4E6F-89EC-58E7D67BF6CB
Electroneuria: urn:lsid:zoobank.org:act:3A4CA23C-2D23-4F8E-9492-93C490AE736C
*E. ronwoodi*: urn:lsid:zoobank.org:act:A52885BB-E2CF-49E8-871B-C4CE8577002C
*Largusoperla charliewattsi*: urn:lsid:zoobank.org:act:43BA0BA8-7818-47FE-BA37-E38F140A457C
*L. billwymani*: urn:lsid:zoobank.org:act:4AB77845-FAB5-4440-8496-935CBD1249B0
*L. micktaylori*: urn:lsid:zoobank.org:act:17E6ED82-03C8-4824-9176-51C34EFDF66F
*L. brianjonesi*: urn:lsid:zoobank.org:act:3FF737E9-D935-48D7-901A-A1F5FBF61CE9.

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
