# Peer review of "‘Rolling’ stoneflies (Insecta: Plecoptera) from mid-Cretaceous Burmese amber"

_PeerJ, doi:10.7717/peerj.5354_

## Round 0.1 · original submission · Minor Revisions

Dear Dr. Sroka and colleagues:

Thanks for submitting your manuscript to PeerJ. I have now received three independent reviews of your work, and as you will see, all are very favorable. Well done! Nonetheless, the reviewers raised some relatively minor concerns about the research, and areas where the manuscript can be improved. Please consider missing references, as well as the treatment of plesiomorphies versus autapomorphies. I agree with the reviewers, and thus feel that their concerns should be adequately addressed before moving forward.

Therefore, I am recommending that you revise your manuscript accordingly, taking into account all of the issues raised by the reviewers. I do believe that your manuscript will be ready for publication once these issues are addressed.

Good luck with your revision,

-joe

·

Basic reporting

No comment

Experimental design

No comment

Validity of the findings

No comment

Additional comments

The only point that may raise doubts is the naming of a new Acroneurinae genus on the basis of larva alone, since larvae are unknown for several extant Acroneuriinae genera. But of course, it is very improbable that Electroneuria ronwoodi would belongs to a recent genus, and I agree that it is probably not a Largusoperla larva. However, it should be corrected that correlation between the length of cerci in larvae and adults regards only to Systellognatha and not all the extant stoneflies.

Reviewer 2 ·

Basic reporting

line 24, you should add the reference to Wang et al too (Fossil record of stem groups
employed in evaluating the
chronogram of insects (Arthropoda:
Hexapoda)

line 25 or 34, there are more recent papers on the phylgeny inside the order
(chen et al 2018
Molecular phylogeny of Systellognatha (Plecoptera: Arctoperlaria) inferred from mitochondrial genome sequences
By: Chen, Zhi-Teng; Zhao, Meng-Yuan; Xu, Cheng; et al.
INTERNATIONAL JOURNAL OF BIOLOGICAL MACROMOLECULES Volume: 111 Pages: 542-547

line 37-38, what do you think about this taxon, there are few supporting synapomorphies, even none is indicated, if I am not wrong

line 82: the nomenclature should have a reference, please precise
line 105, no it is not, the oldest resins are Carboniferous !!

line 128: please precise in the diagnosis which characters are autapomorphies of the new family, it is very important for the readers

line 690, remove one 'is'

line 699: if it is a an ancient trait, it is a plesiomorphy?, so it supports nothing at all
furthermore the basal position of RP is quite frequent in polyneoptera

line 711: no, a plesiomorphy supports nothing at all, if you want to exclude from the crowngroup, maybe find autapomorphies

maybe a key to species could be welcome ?

Experimental design

methods, investigations, research question are ok

Validity of the findings

of interest because it is a new family and oldest taxa in recent families, from a crucial period of changes in aquatic ecosystems

Additional comments

a very nice paper, well written,
maybe just to add some precisions on the apomorphic characters supporting the taxa (in diagnoses) and some small questions to answer

Reviewer 3 ·

Basic reporting

"“Rolling” Stoneflies (Insecta: Plecoptera) from Mid-Cretaceous Burmese amber" (#27907) by Sroka et al., corresponding author Arnold Staniczek;
The paper deals with new species of fossile stoneflies and describes seven new species from four genera. Additionally one new family and three new genera are established. The ms is clearly structured and well written in appropriate English language and fits within the scope of the journal. Descriptions are detailed, although sometimes – due to general conditions of types – quite vague. Figures are of high quality and of high relevance.
As I am not a specialist in fossile stoneflies, I can only suspect that this is state of the art within this specific and sometimes speculative discipline. On the other hand the discussion is deep and pros and cons are carefully debated in a scientific way including morphology, temporal and spatial scales. The systematic position of the newly described taxa are transparently deduced, including some hardly testable hypotheses.
The ms follows a charming approach, naming fossils after „fossile“ rock-band members, which, although a general trend nowadays, dusts the scientific community a little bit.
Summarizing, not being a specialist I can only annotate few misspellings. From my general understanding I support publishing the ms with minor revisions and hope the other reviewers have more substantial comments.
Introduction, page 1: line 20, Stoneflies have aquatic larvae, reformulate: Stoneflies have in general aquatic larvae (there are several semiterrestrial species);
Line 25: within the order was analysed by (Zwick, 1973, 1980): change into: was analysed by Zwick (1973, 1980);
Petroperla mickjaggeri sp. nov. Description, line 185: of femora and on dorsal surface of tibiae: insert a dot after sentence.
After the description I miss a short paragraph dealing with affinities of the family/genus which is given in the discussion but is added in other examples directly below descriptions, please harmonise
Line 374: skip underscore after abdomen
Line 638: start with capital letter after the dot.
Line 690: skip one „is“ within th sentence
Line 701: after e.g. recent instead Recent
Line 710: change prescnce into presence
Line 757: space between Figs and 5D, harmonise Figs and figs within the entire document
Line 785: change Stark and Lenz 1992 in Stark & Lenz, 1992
The abreviation approx. should be changed in approximately within the entire document

Experimental design

there are no experiments

Validity of the findings

New species are a hypotheses which will be validated in future

Additional comments

"“Rolling” Stoneflies (Insecta: Plecoptera) from Mid-Cretaceous Burmese amber" (#27907) by Sroka et al., corresponding author Arnold Staniczek;
The paper deals with new species of fossile stoneflies and describes seven new species from four genera. Additionally one new family and three new genera are established. The ms is clearly structured and well written in appropriate English language and fits within the scope of the journal. Descriptions are detailed, although sometimes – due to general conditions of types – quite vague. Figures are of high quality and of high relevance.
As I am not a specialist in fossile stoneflies, I can only suspect that this is state of the art within this specific and sometimes speculative discipline. On the other hand the discussion is deep and pros and cons are carefully debated in a scientific way including morphology, temporal and spatial scales. The systematic position of the newly described taxa are transparently deduced, including some hardly testable hypotheses.
The ms follows a charming approach, naming fossils after „fossile“ rock-band members, which, although a general trend nowadays, dusts the scientific community a little bit.
Summarizing, not being a specialist I can only annotate few misspellings. From my general understanding I support publishing the ms with minor revisions and hope the other reviewers have more substantial comments.

---

## Round 0.2 · Minor Revisions

Dear Dr. Sroka and colleagues:

Thanks for further revising your manuscript based on the minor concerns raised by the reviewers. I have carefully evaluated it and now believe that your manuscript is almost suitable for publication. After discussion with the Section Editors for this field, and after communication with yourselves, we simply request that you make a remaining edit to remove the additional specimens held in private collections from text and remove the corresponding figures as that material does not adhere to our policies for public deposition of specimens.

Once you have done this, I expect to Accept the submission. I look forward to seeing this work in print, and I anticipate it being an important resource for the entomology and evolutionary biology fields. Thanks again for choosing PeerJ to publish such important work.

Best,

-joe

---

## Round 0.3 · accepted · Accept

Dear Dr. Sroka and colleagues:

Thanks, once again, for further revising your manuscript and accounting for all of the analyzed specimens. I now believe that your manuscript is suitable for publication. Congratulations! It’s All Over Now! I look forward to seeing this work in print, and I anticipate it being an important resource for the entomology and evolutionary biology fields, and also for Rock-n-Roll. Thanks again for choosing PeerJ to publish such important work.

Best,

-joe

#